# Assessment of climate change impact on the malaria vector *Anopheles hyrcanus*, West Nile disease, and incidence of melanoma in the Vojvodina Province (Serbia) using data from a regional climate model

Dragutin T. Mihailović[1], Dušan Petrić[2]*, Tamaš Petrović[3], Ivana Hrnjaković-Cvjetković[4,5], Vladimir Djurdjevic[6], Emilija Nikolić-Đorić[7], Ilija Arsenić[1], Mina Petrić[8,9,10]*, Gordan Mimić[11], Aleksandra Ignjatović-Ćupina[2]

1 Department of Field and Vegetable Crops, Faculty of Agriculture, University of Novi Sad, Novi Sad, Serbia, 2 Department of Plant and Environment Protection, Faculty of Agriculture, University of Novi Sad, Novi Sad, Serbia, 3 Department for virology, Scientific Veterinary Institute "Novi Sad", Novi Sad, Serbia, 4 Institute of Public Health of Vojvodina, Novi Sad, Serbia, 5 Faculty of Medicine, University of Novi Sad, Novi Sad, Serbia, 6 Institute of Meteorology, Faculty of Physics, University of Belgrade, Belgrade, Serbia, 7 Department of Agricultural Economics, Faculty of Agriculture, University of Novi Sad, Novi Sad, Serbia, 8 Avia-GIS NV, Zoersel, Belgium, 9 Department of Physics, Faculty of Sciences, University of Novi Sad, Novi Sad, Serbia, 10 Department of Physics and Astronomy, Faculty of Sciences, University of Gent, Gent, Belgium, 11 BioSense Institute, University of Novi Sad, Novi Sad, Serbia

* mpetric@avia-gis.com (MP); dusanp@polj.uns.ac.rs (DP)

## Abstract

Motivated by the One Health paradigm, we found the expected changes in temperature and UV radiation (UVR) to be a common trigger for enhancing the risk that viruses, vectors, and diseases pose to human and animal health. We compared data from the mosquito field collections and medical studies with regional climate model projections to examine the impact of climate change on the spreading of one malaria vector, the circulation of West Nile virus (WNV), and the incidence of melanoma. We analysed data obtained from ten selected years of standardised mosquito vector sampling with 219 unique location-year combinations, and 10 years of melanoma incidence. Trends in the observed data were compared to the climatic variables obtained by the coupled regional Eta Belgrade University and Princeton Ocean Model for the period 1961–2015 using the A1B scenario, and the expected changes up to 2030 were presented. Spreading and relative abundance of *Anopheles hyrcanus* was positively correlated with the trend of the mean annual temperature. We anticipated a nearly twofold increase in the number of invaded sites up to 2030. The frequency of WNV detections in *Culex pipiens* was significantly correlated to overwintering temperature averages and seasonal relative humidity at the sampling sites. Regression model projects a twofold increase in the incidence of WNV positive *Cx. pipiens* for a rise of 0.5°C in overwintering $T_{October–April}$ temperatures. The projected increase of 56% in the number of days with $T_{max} \geq 30°C$ (Hot Days—HD) and UVR doses (up to 1.2%) corresponds to an increasing trend in melanoma incidence. Simulations of the Pannonian countries climate anticipate warmer and drier conditions with possible dominance of temperature and number of HD

**Data Availability Statement:** All relevant data are within the manuscript and its Supporting Information files.

**Funding:** This paper was realised as a part of the projects "Studying climate change and its influence on the environment: impacts, adaptation and mitigation" (III43007 - DTM, DP, IHC, VDj, ENĐ, IA, GM, AIĆ) and TR31084 (DP, TP, AIĆ) financed by the Ministry of Education and Science, Republic of Serbia (http://www.mpn.gov.rs/). Historical data for mosquito vectors are the outputs of projects supported by the Veterinary Directorate, Ministry of Agriculture and Environment Protection, Republic of Serbia (http://www.minpolj.gov.rs/), and Provincial Secretariat for Science and Technological Development, the Autonomous Province of Vojvodina, project no. 114-451-2142/ 2011 (DP, TP, IHC, AIĆ - http://apv-visokoobrazovanje.vojvodina.gov.rs/). The funders had no role in study design, data collection and analysis, decision to publish, or preparation of the manuscript. MP is affiliated to Avia-GIS NV, which did not play a role in the study design, data collection and analysis, decision to publish, or preparation of the manuscript. Role of Funders Avia-GIS NV provided support in the form of salaries for author (MP) but did not have any additional role in the study design, data collection and analysis, decision to publish, or preparation of the manuscript. The specific roles of the author (MP) are articulated in the 'author contributions' section.

**Competing interests:** All authors except MP have declared that no competing interests exist. Commercial affiliation of MP is to Avia-GIS NV (commercial funder) in which she is under an employment contract since September 2016. Avia-GIS NV provided support in the form of salaries for author (MP) but did not have any additional role in the study design, data collection and analysis, decision to publish, or preparation of the manuscript. This does not alter our adherence to PLOS ONE policies on sharing data and materials.

over other ecological factors. These signal the importance of monitoring the changes to the preparedness of mitigating the risk of vector-borne diseases and melanoma.

# Introduction

Climate change is referred to as "the biggest global health threat of the 21st century" [1].The analysis of outputs from all general circulation models (GCM) suggests that the countries of the Pannonian Plain, including Serbia, are facing significant impacts of climate change, affecting all aspects of human life [2]. The authors of the manuscript (meteorology, entomology, veterinary medicine, and public health experts), have been working together since 2003, promoting the idea of multisectoral collaboration before the One Health Concept was officially inaugurated in the USA in 2007 [3], and endorsed by the European Union (EU) [4] as well as prominent organizations such as the World Health Organization (WHO), Food and Agriculture Organization (FAO), and the World Organization for Animal Health (OIE) in 2018 [5].

In this paper, the authors collected and analysed observed data over 31 years and related a subset to outputs from a Regional Climate Model (RCM). Vector-borne diseases and melanoma are significant climate-driven threats for which risk sources can be clearly defined [6]. Moreover, both present progressively growing environmental threats to the animal as well as human health in the countries of the Pannonian Plane [7,8,9].

The biology and distribution of mosquito vectors and their capacity to transmit mosquito-borne diseases are dependent on many factors such as global trade and travel, urbanisation, habitat destruction, pesticide application, host density, and climate. *Anopheles hyrcanus* and *Culex pipiens* are mosquito species that are vectors of malaria and West Nile virus (WNV) disease, respectively, the two vector-borne diseases distributed worldwide [10,11]. Malaria was eradicated from Serbia and other Balkan states during the last century. However, the spreading of its vectors (*Anopheles* mosquitoes) and the re-emergence of the disease in Greece [12] pose a threat to the South East and Central Europe once again. In 2018, Serbia was the second European country (after Italy) most affected by WNV disease (415 reported cases with 35 fatal outcomes). In Europe, the total number of reported human autochthonous WNV infections in 2018 (n = 2,083) exceeded, by far, the total number from the previous seven years (n = 1,832). During the same transmission season, outbreaks of West Nile fever among equids increased by 30% compared to the number of outbreaks in 2017. In total, 285 outbreaks among equids were reported by the EU Member States in 2018 [13]. Current evidence suggests that inter-annual and inter-decadal climate variability have a direct influence on the epidemiology of vector-borne diseases, with temperature and relative humidity as the principal abiotic factors influencing the life-cycles of the mosquito vector, the pathogen, the host and the interactions between them [14,15].

Melanoma is a malignant disease that has experienced a significant increase in incidence during the last few decades all over the world [16]. Climate change impact on melanoma should be considered as a synergy of changes in UV radiation (UVR) due to stratospheric ozone depletion and the long-term increase of air temperature leading to more prolonged exposure of individuals to UVR doses and consequently to a higher risk of melanoma [17]. Melanoma mortality in the Vojvodina Province (northern Serbia) (VPS) within the period 1985–2004 shows an evident increase, placing it amongst the most vulnerable regions in the world. Thus, Jovanović et al. [7] estimated and made the list of mortality rates from malignant melanoma for males (age-standardised rate/100,000) in Europe (39 countries) for the year

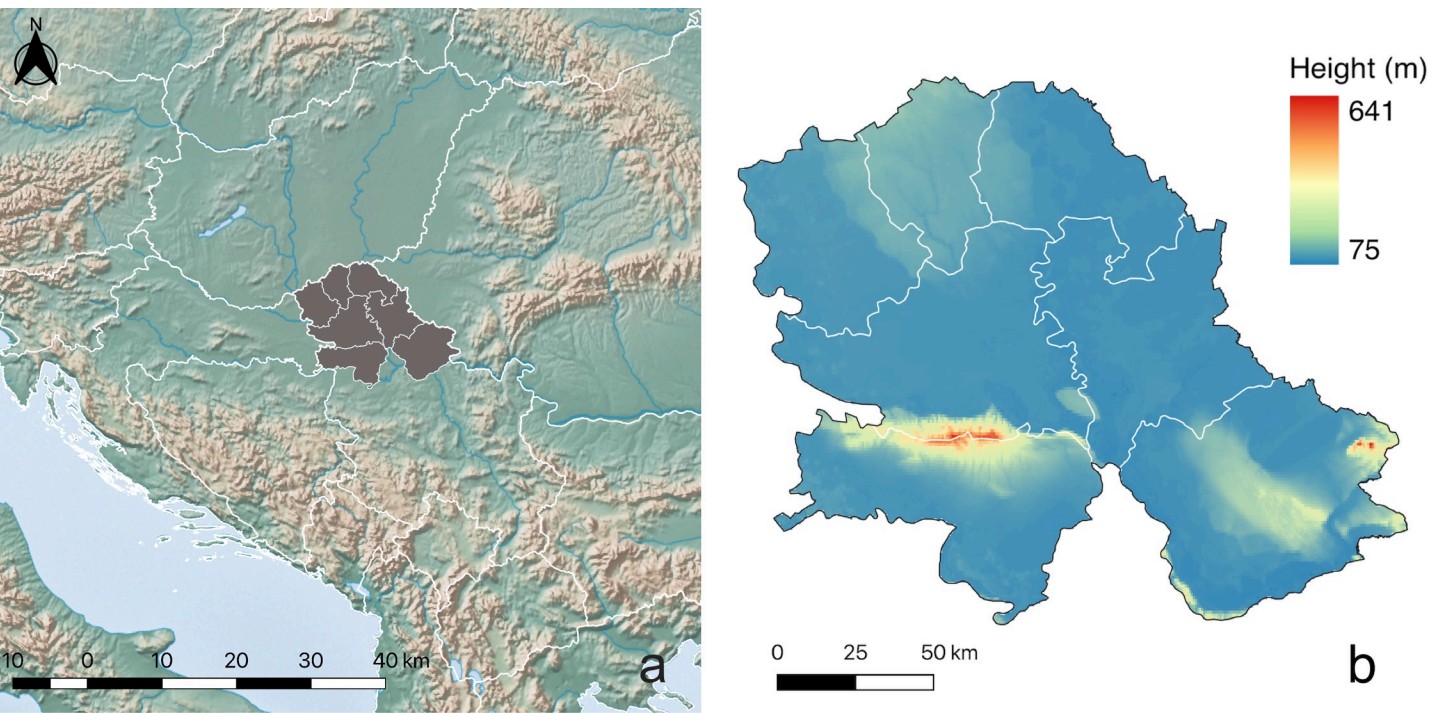

**Fig 1.** (a) Location of the Vojvodina Province (Serbia) in Europe and (b) altitude map. (Made with Natural Earth—naturalearthdata.com).

2000, using European Network of Cancer Registries (ENCR) data. This list shows that the VPS is among the top eleven states (six of them have parts in the Pannonian Plane) listed as the most endangered.

In this study, devoted to revealing the potential impact of climate change on animal and human health, we compared a considerable amount of previously unpublished ecological data obtained from the field and clinical surveys with climate change projections for the VPS, which is representative of the Central European low-altitude areas with a human-dominated landscape (Fig 1). We examined the effects of temperature on the spread and relative abundance of the malaria vector *An*. *hyrcanus* and the "microclimate" differentiation between sites with a specific frequency of WNV occurrence in *Cx*. *pipiens*. We also evaluated the impact of climate change on melanoma incidence as a synergy of changes in UVR doses and the long-term increase in the number of hot days (HD), with daily maximum temperature $\geq 30°C$ using the Eta Belgrade University and Princeton Ocean Model (EBU-POM) regional model data.

## Materials and methods

For the assessment of the climate change and the impact of UVR doses, we used the climatic variables obtained by the coupled regional EBU-POM model for the historical period 1961–2000 and the period 2001–2030 according to the A1B scenario defined in Special Report on Emissions Scenarios (SRES) [18], from now on SRES-A1B. SRES scenarios, which defined future global greenhouse gases emissions, were extensively used in the Intergovernmental Panel on Climate Change (IPCC) Third, and Fourth Assessment Reports. The main storyline behind the A1B scenario is rapid economic growth, followed by a significant increase in greenhouse gases concentrations in the future. In the Fifth Assessment Report (AR5), the Representative Concentration Pathways (RCPs) are introduced, which are possible future

concentration pathways without any storyline behind them [19]. Comparing SRES-A1B and RCPs in terms of the greenhouse gases concentrations, at the end of this century SRES-A1B is the closest to RCP6.0, but for the time horizon used in this study, up to 2030, the difference between any SRES or RCPs are relatively small.

## Study area and climate

The VPS is situated in the northern part of Serbia and the southern part of the Pannonian lowland (18˚51′–21˚33′E, 44˚37′–46˚11′N and 75–641 m.a.s.l. (with the Fruška Gora Mountain in the south) as it is seen in Fig 1A and 1B). This region is the essential food production area in Serbia with a total surface area of 21,500 $km^2$ and a population of about 2 million. This region has a continental climate, with elements of a sub-humid and warm climate (Cfwbx" according to Köppen classification [20]).

## Models and formula used

**The global and regional climate model.** For climate simulations in this study, we used results of the EBU-POM model runs for the SRES-A1B scenario integrated over the period 2001–2030 [21]. The EBU-POM is a two-way, coupled RCM. The atmospheric part is the Eta/National Centres for Environmental Prediction (NCEP) limited area model (resolution 0.25˚ × 0.25˚ on 32 vertical levels; centred at 41.5˚ N, 15˚ E, with boundaries at ±19.9˚ W–E and ±13.0˚ S–N), while the oceanic part is the POM (resolution 0.20˚ × 0.20˚ on 21 vertical levels). The driving global circulation model (GCM) was the ECHAM5 model [22] coupled with the Max Planck Institute Ocean Model (MPI-OM) [23]. More details about model integrations and performed bias correction for VPS can be found in the paper by Mihailović et al. [2]. The POM model was set over the Mediterranean Sea without the Black Sea; for other open seas, the sea surface temperature from the GCM was used as a bottom boundary condition.

## Environmental sampling

**Mosquito vectors.** We used standardised protocols to measure mosquito presence/absence, density, and infestation by WNV. Data were extracted from dry ice-baited trap samples, collected over 31 years at 166 different sites (745 sampled locations, S1 Table) in the VPS, to infer on the trends of local vector status and virus circulation in mosquitoes. In all years, mosquitoes were sampled from May to September, with different spatial intensity and time-frequency governed by the scale and scope of different research projects. For comparison with climate variables, we extracted data obtained in 10 years (1985–6, 2004–5 and 2010–5) for which a standardised surveillance protocol was in place. These periods have the highest number of particular location-year combinations (S1–S3 Tables).

Samples were collected by two different types of dry-ice baited suction traps. During 1985 and 1986 [24,25] by the miniature CDC light trap (CDC) and for 2004 and 2015 by the NS2 trap (our design of dry ice-baited suction trap without light). Both traps were operating without a light source (incandescent light proved not to be attractive/repellent for most mosquito species inhabiting the VPS [25]). The CDC trap has 3–5 times stronger suction power (operated by a 9V battery) than NS2 (operated by 3 x 1.2V batteries), meaning that the increase in density of species observed after 1986 could not be attributed to the change of the type of trap. Traps were operated from the afternoon until the morning of the next day (one trap night), with different periodicity. The specific location of the trap at each site was chosen by experienced entomologists to stabilise variation of the collected data.

We used three parameters to indicate *An. hyrcanus* spread and population growth in the period 1985–2015: i) the ratio of positive to total mosquito samplings per year; ii) the number

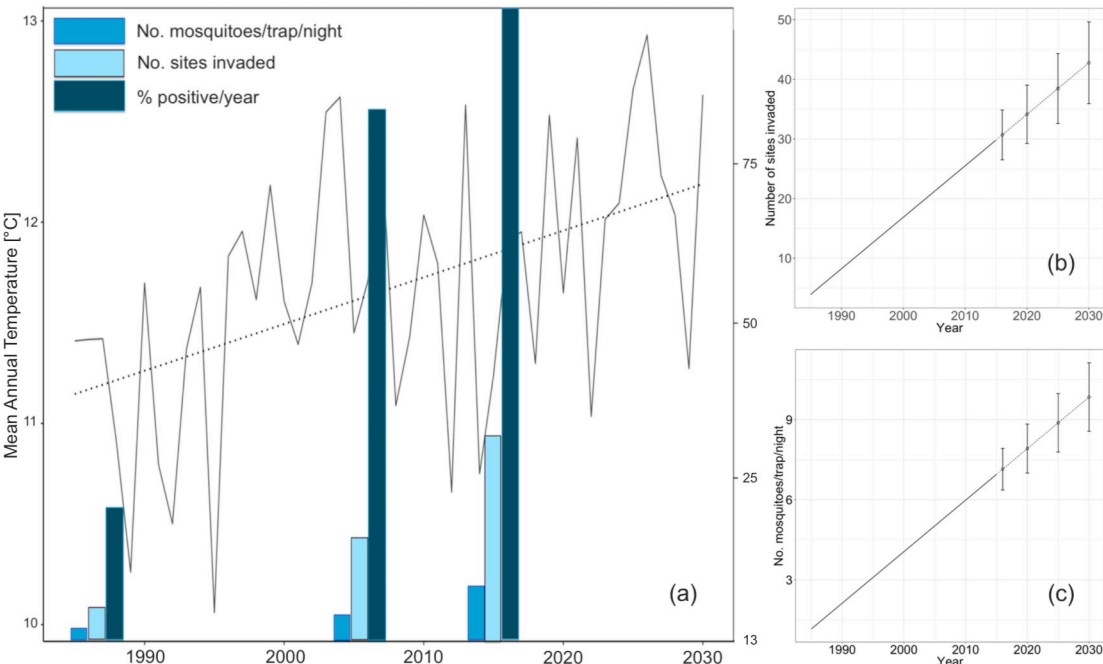

**Fig 2.** (a) The regional climate model EBU-POM projection of the mean annual air temperature ($T_a$) for the period 1985–2030 and: i) number of specimens sampled in one trap during single sampling period (blue columns); ii) the number of sites invaded by *An. hyrcanus* (light blue columns); and iii) relative number of positive samplings per year (dark blue columns), (b) projected increase in the number of sites invaded by *An. hyrcanus* (the period 2001–2030 ±S.E.), and (c) projected increase in the number of the specimens sampled in one trap during single sampling period (2001–2030 ±S.E.).

of sites invaded (positive places where it was looked for, but was not found in the preceding sampling period, and the number of sites where was observed in both periods, i.e. established); and iii) the average number of specimens sampled in one trap during single sampling period from the afternoon of the starting day to the morning of the next day (Fig 2A and 2C). Here, we used data from 1,073 mosquito samples (1985–6, 2004–5 and 2014–5), obtained at 54 location over six years (142 unique location-year combinations) (S2 Table).

For *Cx. pipiens*, the period starting with the first detection of WNV in mosquitoes in Serbia, in 2010 [26], to 2015 was considered. To investigate the impact of microclimate on the complex interaction between *Cx. pipiens* and WNV, we used the following climatic parameters from the EBU-POM model outputs (covering the period 2006–2015) for 11 sites (GPS coordinates–S3 Table) in the VPS with different histories of WNV circulation: (i) mean annual temperature ($T_a$); (ii) overwintering temperature ($T_{OA}$) for the period October–April; (iii) seasonal temperature ($T_{MS}$) for the period May–September; and (iv) relative humidity ($R_{MS}$) for the period May–September. For these sites, we examined the correlation between the frequency of WNV detections in *Cx. pipiens* at each site (from 2010 to 2015) and the corresponding period averages of climate time series for the same site. For detection of WNV, specimens were sampled, anaesthetised by dry ice, identified to species level [27] on dry ice-cooled paper, pooled according to date, location, sex, and species, transported on dry ice to the laboratory, and stored at -70°C before virus detection. Pool size did not exceed 50 mosquito specimens per pool. Mosquito pools were tested for WNV RNA presence by TaqMan-based one-step reverse transcription real-time PCR (RT-qPCR) that amplified both lineage 1 and 2 strains. Viral RNA was extracted using the commercial *ISOLATE II RNA Mini Kit* (Bioline, The Netherlands) according to the manufacturer's instruction. One-step RT-qPCR was conducted

using the commercial kit *RNA UltraSense™ One-Step qRT-PCR System* (Life Technologies Corporation) with the primers and probe that targeted the nucleocapsid protein C gene regions of WNV, as described by Petrović et al. [8]. We analysed the yearly occurrence of the WNV positive *Cx. pipiens* mosquitoes sampled by dry ice-baited traps in the years 2010–2015 across 66 unique location-year combinations (S3 Table). Only traps positioned precisely at the same spot over the entire six-year period are considered for analysis. Numbers allocated to different places (Fig 3) indicate the number of years in the period 2010–2015 in which WNV was detected in sampled *Cx. pipiens* mosquitoes; e.g. 5 indicates that WNV positive *Cx. pipiens* were detected in five out of the six years in the samples collected from the same spot.

**Melanoma incidence.** In the analysis we have used two indicators: (i) melanoma incidence rate that is a measure of the number of new cases ("incidence") per unit of time ("rate") and (ii) cumulative incidence ("incidence proportion" that measures the number of new cases per person in the population over a defined period of time–often called risk or proportion). Melanoma incidence rate (per 100,000 people) for ten years 1995–2004 was based on the data obtained from the paper by Jovanović et al. [7]. From these data, we calculated the cumulative incidence [28]. We have used the model simulation to study the expected impact of climate change on UVR exposure of human skin for nine sites in VPS [PA (Palić), SO (Sombor), KI (Kikinda), NS (Novi Sad), BC (Bečej), ZR (Zrenjanin), SM (Sremska Mitrovica), BK (Bantaski Karlovac), and BG (Beograd)]. For calculating the daily doses of UVR (UVRD) in the study area sites, we have used the following empirical formula $UVRD = 0.002507 \times G_d - 5.985$ (kJ/m2) derived by Malinović-Milićević et al. [29], where $G_d$ is the daily sum of global solar radiation. First, we calculated UVRD from global radiation model outputs using the empirical formula for the nine sites for the period April–September, and then we found the relative change R(UVRD) of those doses as $R(UVRD) = (UVRD-UVRD_k)/ UVRD_k$ where $UVRD_k$ is the dose for 1961–1990 reference period, while the UVRD is calculated for the period 2001–2030.

## Statistics

We considered the papers by Mihailović et al. [2,30] in which Kolmogorov complexity measures [Kolmogorov complexity (KC), Kolmogorov complexity spectrum (KC spectrum), and

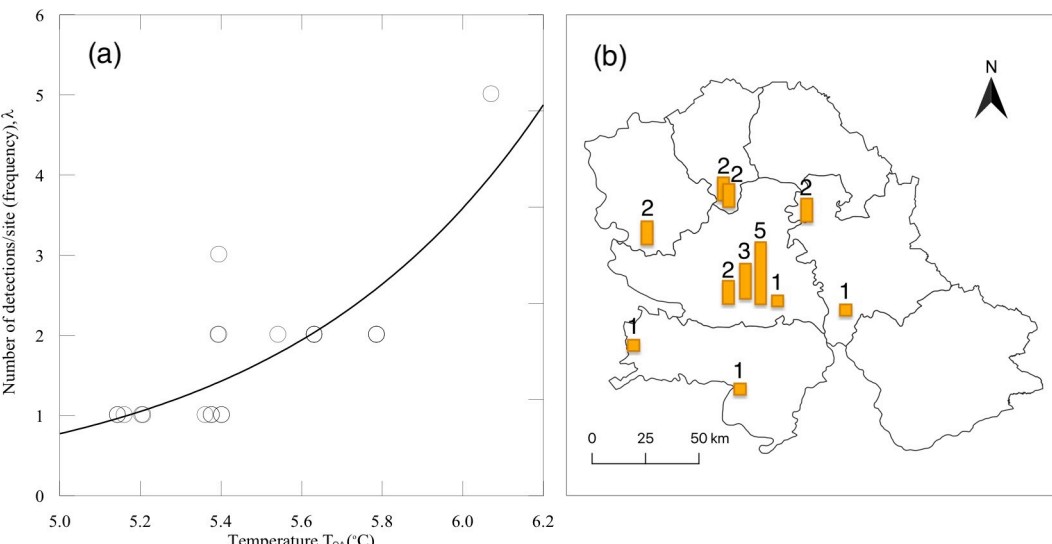

**Fig 3.** (a) Dependence of frequencies (λ) of WNV positive *Culex pipiens* detections at the same site on overwintering temperatures ($T_{OA}$); (b) Frequency of sampling of WNV infected mosquitoes (1–5 times) during six years (bars and numbers) in NUTS3 (Nomenclature of Territorial Units for Statistics) units of the Vojvodina Province, Serbia.

the highest value of the KC spectrum (KCM)], and sample entropy (SE) [31] were used to quantify the regularity and complexity of air temperature and precipitation time series, obtained by the EBU-POM model, representing both deterministic chaos and stochastic processes. We considered the complexity of the EBU-POM model using the observed and modelled time series of temperature and precipitation. We computed the KC spectrum, KC, KCM and SE values for temperature and precipitation. The calculations were performed for the entire time interval 1961–1990: (i) on a daily basis with a size of $n$ = 10,958 samples for temperature and (ii) on a monthly basis with a size $n$ = 360 for the precipitation. The simulated time series of temperature and precipitation were obtained by the EBU-POM model for the given period. The observed time series of temperature and precipitations for two stations: Sombor (SO) (88 m.a.s.l.) and Novi Sad (NS) (84 m.a.s.l.) in the considered area, were taken from daily meteorological reports of the Republic Hydrometeorological Service of Serbia. For both sites, the modelled complexity is lower than the observed one, but with the reliability which is in the interval values allowed by the information measures (KC, KCM, and SE) [32,33,34]. The term model reliability we have used in the following context. The Lyapunov exponent (LLE) relates to the predictability of measured time series, which includes deterministic chaos as an inherent component. Model predictability is here understood as the degree to which a correct prediction of a system's state can be made either qualitatively or quantitatively. In a stochastic analysis, a random process is considered predictable if it is possible to infer the next state from previous observations. In many models, however, randomness is a phenomenon which "spoils" predictability [35]. Deterministic chaos does not mechanically denote total predictability but means that at least it improves the prognostic power. In contrast, stochastic trajectories cannot be projected into the future. If LLE>1 then time series is not chaotic, but is rather stochastic, and predictions cannot be based on chaos theory. However, if $0 < LLE < 1$ it indicates the existence of chaos in time series. In that case, one can compute the approximate time [often called Lyapunov time (LT)] limit for which accurate prediction for a chaotic system is a function of LLE. It designates a period when a specific process (physical, mechanical, hydrological, quantum, or even biological) moves beyond the bounds of precise (or probabilistic) predictability and enters a chaotic mode. According to Frison and Abarbanel [36] that time can be calculated as LLE $\Delta t_{lyap}$ = 1/LLE. If LLE→0, implying that $\Delta t_{lyap} \to \infty$, then long-term accurate predictions are possible. However, many climate time series are highly complex. Therefore, $\Delta t_{lyap}$ can be corrected for randomness in the following way. Similar to $\Delta t_{lyap}$ we can introduce a randomness time $\Delta t_{rand}$ = 1/KC (in time units, second, hour or day). Henceforth, we shall denote this quantity Kolmogorov time (KT), as it quantifies the period beyond which randomness significantly influences predictability. Then, the Lyapunov time corrected for randomness is defined as $[0,\Delta t_{lyap}] \cap [0,\Delta t_{rand}]$. It can be stated that the KT designates the size of the time window within time series where complexity remains nearly unchanged. These findings mean that the models with a KC (and KCM) complexity lower than the measured time series complexity cannot always reconstruct some of the structures contained in the observed data. However, it does not mean that outputs from EBU-POM model do not correctly simulate climate elements since both sites' values indicate the absence of stochastic influences, providing reliable projections of the climate elements [37]. Therefore, the EBU-POM model can be considered as a model having high performances and reliability in projection of temperature and precipitation, two of the most important elements used in different climate research.

For *An. hyrcanus*, the temperature trend was evaluated by the Mann-Kendall test using the R statistical package [38]. Field observed values on species distribution and density for the period 1985–2015 and forecasts of the numbers of sites invaded and specimens sampled for the period 2016–2030 based on linear trends were obtained by the Eviews 9.5 software [39].

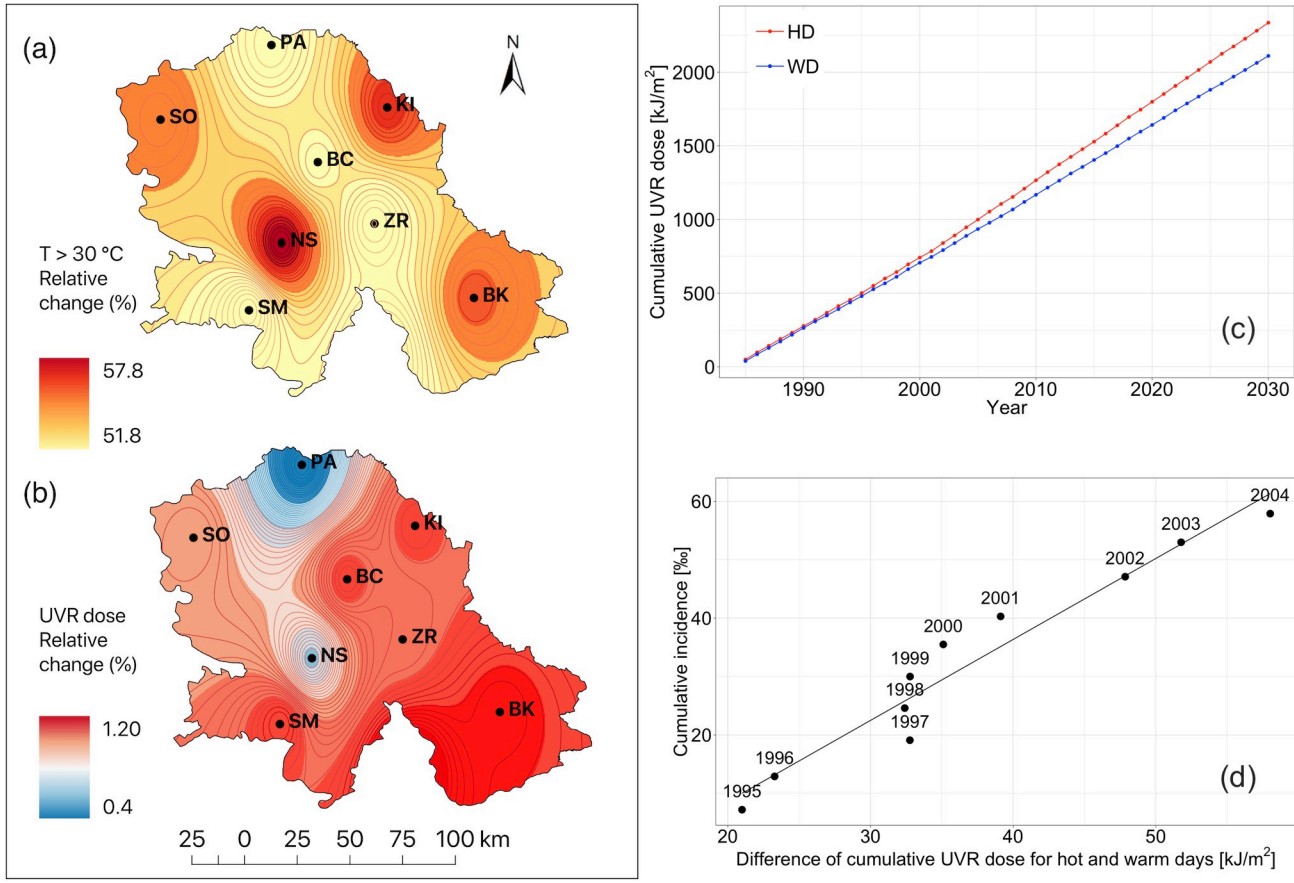

**Fig 4.** (a) Relative change of hot days (HD) and (b) UVR radiation doses [R (UVRD)] for the period 2001–2030 compared to the period 1961–1990; (c) cumulative values of mean UVR doses for the period 1985–2030 (averaged for nine sites: PA, SO, KI, NS, BC, ZR, SM, BK, and BG) under the SRES-A1B scenario (for WD and HD days); and (d) cumulative incidence of melanoma for the period 1995–2004 in the Vojvodina Province, Serbia.

For *Cx. pipiens*, the relationship between yearly frequency of WNV detection in mosquitoes, air temperature, and relative humidity (derived from the climate model) was estimated using Spearman's Rank-Order Correlation and a Poisson regression model (Statistica 13 [40]). For melanoma, the linear regression model was used for modelling the cumulative incidence of melanoma versus the difference of the cumulative UVR doses for hot and warm days (Fig 4D). Analysis of residual distribution was done by Shapiro-Wilk test (Statistica 13 [40])

## Results

### Mosquito vectors

Fig 2A shows an evident linear trend of the mean annual temperature $T_a$ for the period 1985–2030 ($r = 0.467$; $p = 0.001$; $\tau = 0.328$) calculated from the EBU-POM regional model outputs for 29 representative sites in the VPS. All parameters that were chosen for the evaluation of the spread and population increase of *An. hyrcanus* were positively, but to a different extent, correlated to the time argument (periods in which sampling was performed since the beginning of monitoring in 1985) indicating a monotonic trend. The increase of parameters follows the trend of $T_a$ (Fig 2A). The strongest correlation was found for the increase in the ratio of positive samplings ($r = 0.986$; $p = 0.000307$; $\tau = 0.828$), followed by the number of mosquitoes per trap night ($r = 0.919$; $p = 0.009639$; $\tau = 0.733$), and the number of sites invaded ($r = 0.889$;

$p = 0.01766$; $\tau = 0.6$). By 2030 we anticipate a further increase in numbers of invaded sites and adult females sampled, by 1.71 and 1.27 fold, respectively (Fig 2B and 2C).

Spearman rank-order correlation between the frequency of WNV detections in *Cx. pipiens* at 11 sites and the corresponding mean values of climate time series was the highest for $T_{OA}$ ($r = 0.755$; $p = 0.00008$), then for $T_a$ ($r = 0.616$; $p = 0.00294$), $R_{MS}$ ($r = 0.499$; $p = 0.02119$), and $T_{MS}$ ($r = 0.477$; $p = 0.02856$). Fig 3A depicts the Poisson regression model for the dependence of a number of detections per site (frequency - $\lambda$) on $T_{OA}$, which is statistically significant ($p = 0.01393$). The output of the model ($ln\lambda = -7.923 + 1.533 \times T_{OA}$) indicates that for an increase of 0.5˚C in $T_{OA}$ (presuming that all other factors needed for the circulation of WNV are kept constant), a twofold increase in the incidence of WNV positive *Cx. pipiens* could be projected. Fig 3B depicts that most of the sites with the high frequency of WNV occurrence ($\geq 2$) were distributed along the northwest-southeast axis of the VPS.

### Melanoma incidence and UVR doses

Fig 4B shows the positive relative change of UVRD, remarkably covering an eastern, southern, western, and partly central area of VPS. Specifically, the projected increase is twofold going from the west and northwest (0.60%) towards the east and southeast where it reaches values of about 1.20%. The EBU-POM model (for nine sites) shows a significant expected increase of 56% in the number of HD days in the VPS (Fig 4A), compared to the period 1961–1990. Additionally, we observed a decrease of 1.1% in the number of days with maximum air temperature $T_{max} \geq 25$˚C (Warm Days—WD). This prolongs the exposure of outdoor working adults to UVR and thus leads to an increase in melanoma risk. This risk becomes even more significant because of the increase in cumulative values of UVR doses (Fig 4C). Fig 4D depicts the cumulative incidence of melanoma for the period 1985–2004 with an increasing monotonic trend ($r = 0.9712$ and $p = 0.000003$).

## Discussion

Here we presented an intriguing comparison of the impact of climate change on complex systems including mosquito vectors, pathogens, and melanoma, which are all indicators of the risk imposed on human health. Our objectives were to use historical, previously unpublished sets of entomological and published clinical data and examine the importance of temperature in contributing to the spreading of the malaria vector *An. hyrcanus*; to differentiate between sites with a specific frequency of WNV occurrence in *Cx. pipiens*, and to assess the impact of increasing UVR and HD on melanoma incidence using the EBU-POM regional model data. A similar approach was recently used in observing the dramatic decline in total flying insect biomass in protected areas in Germany [41].

Despite globalisation trends, researchers have become "closed" in their ever-smaller communication circles which are not limited by state or national borders but by the professional language and way of thinking. Thus, by the end of the 20th century, the scientific community has been faced with problems in communication within its confines. One of the principal reasons why vector-borne diseases (VBD) are so difficult to predict, is the complex interaction of multiple factors (vector, host, pathogen, environment including short-term weather patterns and long-term climate change) in space and time [42,43]. Only groups from multiple sectors that communicate and work together on specific aspects of VBD systems will be able to answer the most exciting and pressing problems in the field [42]. Authors of this paper started collaboration in 2003 comparing the climates of the foci of WNV circulation in USA (California Central Valley) and Europe (Bucharest area) with VPS. As compared climates showed quite similar patterns, colleagues from public health and veterinary joined the initial group of

meteorologists and medical entomologists. With the idea to better draw upon the resources and insights of the various sectors we designed and implemented research and programmes to achieve better outcomes in the control of zoonoses (diseases that can spread between animals and humans, e.g., WNV disease). This led us to the following achievements: (i) the first detection of WNV in horses in Serbia in 2009 [44]; (ii) the first detection of WNV in mosquitoes in Serbia in 2010 [26]; (iii) the first detection of WNV in wild birds in Serbia in 2012 [45]; (iv) development and implementation of the national programme of WNV surveillance in mosquito, bird and horse population [8], combined with human surveillance in VPS from 2014; (v) increased visibility to ECDC, EFSA and WHO; (vi) the first detection of imported dengue human case in Serbia in 2016 [46]; and (vii) development and implementation of "One Health" programme in VPS from 2018. We are quite sure that much less would have been achieved without multidisciplinary communication and collaboration initiated in 2003, and this paper would not have been compiled.

The temperature trend over the period of observations used in this study and for the future time horizon following A1B scenario obtained with the EBU-POM regional climate model is within the multi-model ensemble (MME) spread of regional climate models with the similar configuration used in the ENSEMBLES project [47]. For the region of interest, temperature change for the period 2001–2030 with respect to the period 1961–1990, in the EBU-POM integration is 0.75˚C and in the case of ENSEMBLES MME spread range is 0.5–1.5˚C [48]. Furthermore, over period 2001–2030, uncertainty in temperature projections, related to the scenarios is just small fraction of total uncertainty [49], due to the fact that there is no significant difference in greenhouse gases concentrations for different SRES scenarios. Consequently, it can be considered that for the period 2001–2030, in case of other scenarios, the estimated range will be similar. Following this findings, even that selection of single model and single scenario does not allow estimate of the full uncertainty in the future, presented results that relay on temperature change, can be seen as plausible future realisation within total uncertainty range, and realistic response of the complex system for further increase in temperature as a forcing driver. In future studies, it will be beneficial to introduce results of other regional climate models, such as multi-model ensemble from the CORDEX framework, as a need for better understanding of different uncertainties in the results.

## Mosquito vectors

Until the end of the 20[th] century, northern Serbia was considered the northern limit for the distribution of *An. hyrcanus* in Europe. The first detection in Serbia dates from 1979 [50] from the north part of VPS. We found it in the central part of the Province in 1985 [25] and since then have been noticing its continued spread. Several records north from Vojvodina, in Slovakia in 2004 [51], the Czech Republic in 2005 [52], and Austria in 2012 [53] confirm our observation. Due to its exophilic and exophagic behaviour, *An. hyrcanus* has never been considered as the primary vector of malaria in Europe. Its spread to higher latitudes, combined with the changes in human behaviour (increased outdoor leisure activities, the mobility of humans, number of seasonal workers in the field, number of migrants in Europe), might increase its vector capacity. The similar northern spread of population distribution range that was registered for *An. maculipennis* s.s. in Russia [54], and *Culiseta longiareolata* in southern (in 2012 [55]) and northern (in 2013 [56]) Austria might well represent the tendency described with our model.

The latest illustration of similar changes is the finding of *Uranotaenia unguiculata*, a thermophilic mosquito species frequently occurring in the Mediterranean basin, in northern Germany, some 300km north of the previous northern limit [57].

During the period 2001–2030 in which the spread and population growth of *An. hyrcanus* is expected, the intensity of UVR is likely to increase in the VPS (Fig 4A). Positive trends which are present in our observations might indicate that the findings on the negative influence of UVR and blue-light radiation (this radiation has a wavelength between approximately 380nm and 500nm; it has a very short wavelength, and so produces a higher amount of energy) on adult mosquitoes under laboratory conditions [58,59] could not be simply translated to the field. This experimental evidence does not mean unavoidably that the blue light radiation has a significant influence on adult mosquitoes in field conditions since they can actively escape over-exposure to radiation.

The WNV transmission cycle involves mosquito vectors and birds, but equines and humans are also susceptible to infection [60,61]. Although WNV infections have been described in a wide variety of vertebrates, birds are the main natural reservoir. Hundreds of wild and domestic avian species have been described as susceptible to WNV infection, but many of these showed only subclinical infection [62]. In horses, WNV infection is also frequently clinically unapparent, but around 10% of cases develop neurological disorders with up to 50% mortality rates [61,63]. An increasing number of severe outbreaks in horses have been reported in Europe in the past decade, including a large one that took place in northeast Italy in 2008 involving 251 stables with 794 cases and five deaths [63].

From the first detection of WNV in 8 out of 81 dead wild birds found in Serbia [45], each year WNV nucleic acid was detected in dead or captured wild birds during summertime [8]. Serological testing of horses sampled during 2009–2010 showed that 12% of 349 horses from the northern part of Serbia had neutralising WNV antibodies [44]. After that, each year horse WNV cases were detected by the positive serological response (IgG and IgM antibody seroconverted horses) [8] or as a clinical manifestation of West Nile neuroinvasive disease [64].

A positive association between WNV disease and temperature was already reported in Europe [15,65] where climate and landscape were critical predictors of WNV disease outbreaks [66]. Our focus was not on the number of human WNV cases, but the suitability of sites/microhabitats with different air temperatures for WNV circulation in mosquitoes, which may well correspond to a higher risk of transmission. We found that sites with higher $T_{OA}$ and $T_a$ were characterized with the higher frequency of WNV presence in mosquitoes. Clustering of cases with an incidence higher than one in six years coincided with an area of a significant grouping of mosquito, bird, horse, and human cases in 2014 and 2015 [9] (Fig 5). This is in concurrence with Tran et al. [67] and Marcantonio et al. [66], who found that average summer temperatures are positively correlated with WNV human incidence. It seems that temperature in semi-urban areas is an essential environmental factor influencing WNV circulation (landscape suitability for reservoir host and mosquito vector, host availability, and precipitation/water availability are somewhat similar in investigated semi-urban areas of VPS), as it affects both mosquito vector abundance [9] and virus replication. Prediction of a two-fold increase in virus incidence for each 0.5°C increase in $T_{OA}$ indicates but does not necessarily means, that the number of human cases could increase too. Therefore, our findings support the statement that climate change is likely to intensify the re-emergence of WNV in Europe [68].

## Melanoma incidence and UVR

According to World Health Organization (WHO) (1992) and many other authors [69,70], exposure to UVR radiation is considered to be a major etiological factor for all three forms of melanoma (i) basal cell carcinoma (BCC), (ii) squamous cell carcinoma (SCC), and (iii) malignant melanoma (MM). We found the correlation between MM and climate changes impact on UVR and also the number of HD. We see the impact as a modification of ambient UVR

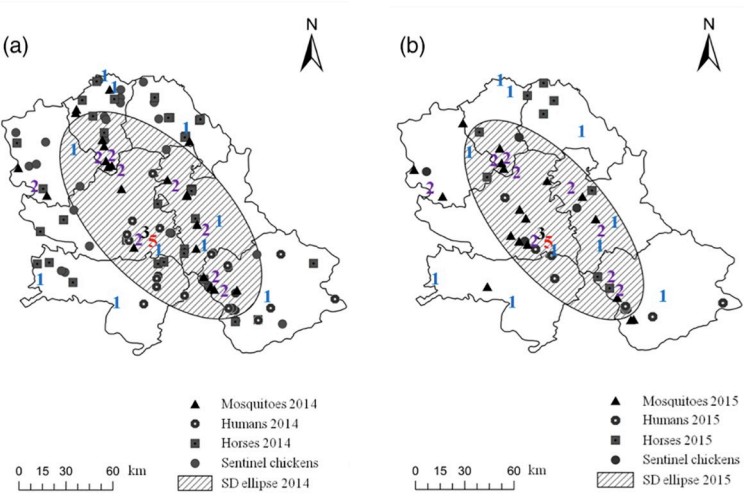

**Fig 5.** Frequency of sampling of WNV infected mosquitoes (1–5 times, coloured numbers) during the period 2010–2016, superimposed over a cluster of mosquito, bird, horse, and human WNV cases in (a) 2014 and (b) 2015. The standard deviation ellipse (1-$\sigma$) is centered around the mean coordinates of the total number of mosquito, bird, horse and human WNV cases (modified after Petrić et al. [9]).

through influences on other variables such as clouds and aerosols. However, that impact might be more pronounced through the impact of changes in outdoor ambient temperature which will influence people's behaviour and increase the time they spend outdoors, i.e. exposure to both higher UVR and higher temperatures [17]. Experiments with animals clearly show that increased temperatures enhance UVR-induced melanoma compared to the room temperature. In an intriguing study, van der Leun [71] postulated that long-term elevation of temperature by 2˚C, as a consequence of climate change, would increase the carcinogenic effects of UVR by 10%. Our results for the UVR in the VPS are generally similar to the ones obtained by Malinović-Milićević et al. [72] and Malinović-Milićević and Radovanović [73], who reported the following changes: (i) the reduction of yearly averages for the total ozone of 3.44% and 3.21%, and (ii) increase in erythemal UVR dose of 6.9% and 9.7% for the periods 1990–1999 and 2000–2009, respectively.

According to Jovanović et al. [7], the incidence rate of MM cancer in VPS for the period 1985–2004 is higher than in central Serbia and is comparable with the majority of the central European countries with the highest melanoma incidence rate in the world [74]. However, most studies do not deal more quantitatively with the relationship between UVR doses and exposure during HD days, and as it has been stated above, the cumulative exposure to sunlight is probably the most critical risk factor for MM and SCC cancers, while BCC is more associated with intensive short-term exposure [75]. Thus, the increasing trend in the number of melanoma incidence in the VPS for the period 1985–2004 (Fig 4D) can be ascribed to (i) the increase in the number of HD days for about 55% and (ii) the increase in cumulative values of UVR doses for the period 1985–2030. In a cohort study, Wu et al. [76] considered the impact of long-term UV radiation flux on skin cancer risk. Comparing with participants in the lowest quintile of cumulative UV radiation flux in adulthood, they found that participants in the highest quintile had multivariable-adjusted risks (cumulative incidence). According to Vandenbroucke and Pearce [28], some studies where cumulative incidence is used can over-represent the trends.

From a statistical point of view, the linear regression model for modelling the cumulative incidence of melanoma versus the difference of the cumulative UVR doses for hot and warm

days (Fig 4D) is acceptable. Parameters are statistically highly significant ($r$ = 0.9712 and $p$ = 0.000003) while analysis of residual distribution shows a good agreement with the normal distribution (Shapiro-Wilk test, $W$ = 0.9608, $p$ = 0.7952).

We hope that our results will indicate the importance of long-term monitoring/surveillance programs for providing crucial data to evidence the ongoing biological alteration triggered by climate change. Nonetheless, it is difficult to say how broadly our data represent the trends elsewhere. We believe that the specificity of the observations offers a unique window into the state of some of the planet's pressing threats to human health. Also, in the case where humans are exposed to UVR, due to the nature of their work (the VPS is an exclusively agricultural area), it is necessary to (i) establish a broader network for UVR measurements and warning centres and (ii) increase the awareness of the melanoma as a result of increased amount of UVR.

## Supporting information

**S1 Table. Overview of dry-ice trap samples sizes.** For each year, the number of locations sampled, the number of location re-sampled, and total number of samples are presented. Exposure time at the trap locations was similar (14 ± 2h).
(XLSX)

**S2 Table. Number of the total trap nights, positive trap nights and *Anopheles hyrcanus* specimens sampled at 54 selected sites in the Vojvodina Province, Serbia during the years 1985–86, 2004–5 and 2014–15.**
(XLSX)

**S3 Table. Frequency of sampling of WNV infected mosquitoes (1–5 times) in the Vojvodina Province, Serbia, during the period 2010–2016.**
(DOCX)

## Acknowledgments

For the collection of samples over the period 1985–2005, we warmly thank Petar Igrić, Dragan Aradski, Dragan Dondur, Vid Srdić, Mihaela Kavran, Slavica Vaselek, and Dubravka Pudar. The works of Dušan Petrić and Mina Petrić were done under the frame of EurNegVec COST Action TD1303.

## Author Contributions

**Conceptualization:** Dragutin T. Mihailović, Dušan Petrić, Tamaš Petrović, Emilija Nikolić-Đorić, Ilija Arsenić, Gordan Mimić.

**Data curation:** Dragutin T. Mihailović, Dušan Petrić, Tamaš Petrović, Ivana Hrnjaković-Cvjetković, Vladimir Djurdjevic, Aleksandra Ignjatović-Ćupina.

**Formal analysis:** Dragutin T. Mihailović, Dušan Petrić, Emilija Nikolić-Đorić, Ilija Arsenić, Mina Petrić, Gordan Mimić.

**Funding acquisition:** Dušan Petrić.

**Investigation:** Dragutin T. Mihailović, Dušan Petrić, Tamaš Petrović, Ivana Hrnjaković-Cvjetković, Emilija Nikolić-Đorić, Mina Petrić, Gordan Mimić.

**Methodology:** Dragutin T. Mihailović, Dušan Petrić, Vladimir Djurdjevic, Emilija Nikolić-Đorić, Ilija Arsenić, Gordan Mimić.

**Project administration:** Dušan Petrić.

**Resources:** Dragutin T. Mihailović, Dušan Petrić, Tamaš Petrović, Aleksandra Ignjatović-Ćupina.

**Supervision:** Dragutin T. Mihailović, Dušan Petrić, Emilija Nikolić-Đorić.

**Validation:** Dragutin T. Mihailović, Dušan Petrić.

**Visualization:** Dragutin T. Mihailović, Dušan Petrić, Mina Petrić.

**Writing – original draft:** Dragutin T. Mihailović, Dušan Petrić, Tamaš Petrović, Ivana Hrnjaković-Cvjetković, Emilija Nikolić-Đorić, Mina Petrić, Gordan Mimić.

**Writing – review & editing:** Dragutin T. Mihailović, Dušan Petrić, Emilija Nikolić-Đorić, Mina Petrić.

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
