## [Decision Letter · Decision Letter 0]

21 Aug 2019

PONE-D-19-16900

Assessment of climate change impact on malaria vectors, West Nile disease, and incidence of melanoma in the Vojvodina Province (Serbia) using data from a regional climate model

PLOS ONE

Dear Ms. Petric,

Thank you very much for submitting your manuscript "Assessment of climate change impact on malaria vectors, West Nile disease, and incidence of melanoma in the Vojvodina Province (Serbia) using data from a regional climate model" (#PONE-D-19-16900) for review by PLOS ONE. As with all papers submitted to the journal, your manuscript was fully evaluated by academic editor (myself) and by independent peer reviewers. The reviewers appreciated the attention to an important health topic, but they raised substantial concerns about the paper that must be addressed before this manuscript can be accurately assessed for meeting the PLOS ONE criteria. Therefore, if you feel these issues can be adequately addressed, we invite you to submit a revised version of the manuscript that addresses the points raised during the review process. We can’t, of course, promise publication at that time.

We would appreciate receiving your revised manuscript by Oct 05 2019 11:59PM. To enhance the reproducibility of your results, we recommend that if applicable you deposit your laboratory protocols in protocols.io, where a protocol can be assigned its own identifier (DOI) such that it can be cited independently in the future. For instructions see: http://journals.plos.org/plosone/s/submission-guidelines#loc-laboratory-protocols

We look forward to receiving your revised manuscript.

Kind regards,

Abdallah M. Samy, PhD

Academic Editor

PLOS ONE

**Journal Requirements:**

**1. Thank you for including your competing interests statement; "**The authors have declared that no competing interests exist."

We note one or more authors are affiliated to Avia-GIS NV.

2. We note that [Figure(s) 1] in your submission contain [map/satellite] images which may be copyrighted. All PLOS content is published under the Creative Commons Attribution License (CC BY 4.0), which means that the manuscript, images, and Supporting Information files will be freely available online, and any third party is permitted to access, download, copy, distribute, and use these materials in any way, even commercially, with proper attribution. For these reasons, we cannot publish previously copyrighted maps or satellite images created using proprietary data, such as Google software (Google Maps, Street View, and Earth). For more information, see our copyright guidelines: http://journals.plos.org/plosone/s/licenses-and-copyright.

1.    You may seek permission from the original copyright holder of Figure(s) [1] to publish the content specifically under the CC BY 4.0 license. 

**Additional Editor Comments:**

I invited and received two reviews for your manuscript. All reviews raised some substantial concerns about your manuscript as it currently stands. As such, I would recommend “major revision”. I would kindly ask you to go through all comments raised by each reviewer and address them properly before sending a revised version of this manuscript. Please check all PLOS ONE style requirements available via https://journals.plos.org/plosone/s/submission-guidelines before submitting the revised version.

**Reviewers' comments:**

Reviewer's Responses to Questions

**Comments to the Author**

1. Is the manuscript technically sound, and do the data support the conclusions?

Reviewer #1: Yes

Reviewer #2: Partly

2. Has the statistical analysis been performed appropriately and rigorously? 

Reviewer #1: Yes

Reviewer #2: N/A

3. Have the authors made all data underlying the findings in their manuscript fully available?

Reviewer #1: Yes

Reviewer #2: Yes

4. Is the manuscript presented in an intelligible fashion and written in standard English?

Reviewer #1: Yes

Reviewer #2: Yes

5. Review Comments to the Author

Reviewer #1: Comments on the manuscript PONE-D-19-16900

Title: Assessment of climate change impact on malaria vectors, West Nile disease, and incidence of melanoma in the Vojvodina Province (Serbia) using data from a regional climate model

Authors: Dragutin T. Mihailović, Dušan Petrić, Tamaš Petrović, Ivana Hrnjaković Cvjetković, Vladimir Đurđević, Emilija Nikolić Đorić, Ilija Arsenić, Mina Petrić, Gordan Mimić

The work presented in paper Mihailović et al. is interesting. The objective of the authors was to compare data from the mosquito field collections and medical studies with regional 29 climate model projections to examine the impact of climate change on the circulation of West Nile virus (WNV), the spreading of the malaria vector Anopheles hyrcanus and the incidence of melanoma. The comparison was done with the coupled regional Eta Belgrade University and Princeton Ocean Model for the period 1961-2015 using the A1B scenario, and the expected changes up to 2030. Overall, significant correlation was found between the frequency of WNV in Culex pipiens and the overwintering temperature averages and seasonal relative humidity at the sampling sites. Correlation was also found between the spreading and relative abundance of Anopheles hyrcanus and the trend of the mean annual temperature. There was also an increase in melanoma incidence.

Minor comments to authors

Title

Authors wrote “malaria vectors” but the only presented data on only one vector Anopheles hyrcanus

Abstract

L32: ……………and 10-years. Delete the hyphen

L37: ……………Culex. pipiens. Delete the dot

Introduction

L49-53: Are the authors referring to themselves when they stated, “The authors (……….), have been working together…….”?

L72: Authors should write, “Climate change……..” instead of “The climate change……..”

L75: Author should write, “Melanoma mortality…………..within the period 1985-2004

L78: Authors should define the acronym ENCR, as this is used here for the first time.

L81-82: Authors should use past tense in the sentence “……we compared considerable of previously……”

L54-55: This sentence is not clear for me. I suggest this: “In this paper, we analysed observed data collected over a period of 31 years……”

Materials and Methods

L96: Authors should define SRES-A1B scenario for the first time.

L115: Authors used only one formula, the subtitle should be in singular

Results

- Authors should specify the exact p-values instead of writing p<0.05 or p>0.05

L207: “The Poisson regression model for the dependence of a number of detections per site (frequency-λ)…………………………is highly significant”. Authors stated it was highly significant, but from my perspective, p<0.05 is not a specific indication of high significance. Could the give the exact value of p?

- Fig2b and 2c are fuzzy

- Fig4. Colors of fig4c are confusing

It will be more interesting if the authors used only vector-borne diseases data in this paper.

N.B: Other comments are incorporated in the manuscript

Reviewer #2: Authors are presenting an interesting paper regarding the effects of climate change in Northern Serbia considering three independent measures: The spread of Anopheles hyrcanus, the presence of West Nile Virus in Culex pipiens, and the incidence of melanoma cases. The paper is interesting, however, discussion should be improved specially on the uncertainty of future predictions since they are using just one climatic model. Further, their results should be stated more carefully since their model rely on assumptions (e.g., manually selected variables) which are also not clearly stated.

Major comments:

The paper is showing results in the order: malaria vector, WNV, and melanoma. I suggest following the same order in the abstract.

Authors are using one of the SRES future climatic scenarios; currently the standard for future climate studies are the RCP scenarios. Authors should describe the nature of the SRES-A1B scenario, which is not mentioned in any part of the study. Further, authors should explicitly discuss uncertainty on their predictions since they are not using other scenarios or other climatic models.

Lines 176-180. There is no discussion or results regarding these sentences. Was the comparison between EBU-POM model and the Republic Hydrometeorological Service of Serbia perfect? What is the implication of this approach on the overall paper?

Line 277-280: There is no evidence in this paper supporting this affirmation since the variables analyzed corresponded to three temperature related variables and just one considering humidity. Moreover, results were never compared statistically; modify accordingly.

Figure 2: Expand the acronym CRCM. Also, double check the legend, which is describing red and green colors but the figure is only showing different shades of blue.

Figure 4: Add WD and HD to the corresponding legend of the graphic. Is there a Croatian sentence in the legend? Please describe how the melanoma incidence was calculated, is the y axis showing incidence or number of cases? Cumulative incidence is known to over-represent trends (see reference: Vandenbroucke & Pearce, 2012, doi: 10.1093/ije/dys142), try to use incidence rate instead.

In table S3 consider adding the number of mosquito samples per site.

Figure 6 can be replaced with the statistics of such graphic for readers’ interpretation.

Authors are justifying the paper under the ‘One Health’ concept, however they are not discussing the idea further. I would like discussing explicitly the benefits of putting together a set of multidisciplinary specialists to the development of the manuscript and how this contribution is part of the one health concept.

Minor comments:

Please use Oxford comma across the manuscript: e.g., Line 30: ‘the malaria vector, and the incidence of melanoma’.

Line 28: Authors never discuss problems related with animal health, thus, I suggest avoiding this kind of affirmations (e.g., line 81).

Line 28: Methods on the paper should be written in past tense: e.g., COMPARED. Review this in the rest of the manuscript, e.g., line 82.

Line 30: ‘the spreading of ONE malaria vector’

Line 37: ‘Culex.’ should be corrected, only Cx.?

Line 40: This is the first time you are mentioning HD, please expand the acronym, review this in the rest of the manuscript, for example EU in line 50, or ENCR in line 78.

Line 44: Specify the risk that you are addressing with this research.

Line 54: extra ‘Collected’ after ‘observed data’, please erase.

Line 55: Add ‘are’ after the word ‘melanoma’ at the end of the sentence.

Line 58: Here you need a reference for the environmental threat represented for the animal and humans at the Pannonian plane.

Line 63: You need a reference for the affirmation of malaria as worldwide detrimental vector-borne disease.

Line 70: Consider adding a reference of how temperature and relative humidity are principal abiotic factors for WNV and An. hyrcanus.

Line 76: Be consistent across the whole manuscript, use either - or – without spaces to separate year timeframes, 1976–2004 is preferred.

Line 102: Add corresponding reference for the Köpen classification.

Line 122: “Data were...”

Line 126: 1985–1986.

Line 158: Briefly describe the method of WNV detection, i.e., RT-PCR or the corresponding one before referencing Petrovic et al 2018.

Line 172: Describe the indicators briefly before referencing Jovanovic et al 2009.

Line 227: Is the formula correct: warm days - WD?

Line 263: Consider changing ‘indicate that the findings supporting’ by ‘support’

Line 273: Authors are not showing incidence rates, just presence of WNV in mosquitoes.

6. PLOS authors have the option to publish the peer review history of their article (what does this mean?). If published, this will include your full peer review and any attached files.

Reviewer #1: Yes: Koffi Mensah Ahadji-Dabla

Reviewer #2: No

---

## [Author Response · Author response to Decision Letter 0]

27 Sep 2019

Dear Dr Samy,

We are pleased to submit the revised version of “Assessment of climate change impact on the malaria vector Anopheles hyrcanus, West Nile disease, and incidence of melanoma in the Vojvodina Province (Serbia) using data from a regional climate model” (#PONE-D-19-16900). We are thankful to the academic editor for time and efforts he invested in providing detailed instructions for changes needed before resubmission.

We corrected the style in order to meet PLOS ONE's style requirements, including those for file naming.

We amended our Competing Interests Statement, and it now reads as follows:

Competing Interests Statement:

All authors except MP have declared that no competing interests exist. Commercial affiliation of MP is to Avia-GIS NV (commercial funder) in which she is under an employment contract since September 2016. Avia-GIS NV provided support in the form of salaries for author (MP) but did not have any additional role in the study design, data collection and analysis, decision to publish, or preparation of the manuscript. This does not alter our adherence to PLOS ONE policies on sharing data and materials.

We amended our Funding statement and Role of Funders, and it now reads as follows:

Funding Statement:

This paper was realised as a part of the projects "Studying climate change and its influence on the environment: impacts, adaptation and mitigation" (III43007 - DTM, DP, IHC, VDj, ENĐ, IA, GM, AIĆ) and TR31084 (DP, TP, AIĆ) financed by the Ministry of Education and Science, Republic of Serbia (http://www.mpn.gov.rs/). Historical data for mosquito vectors are the outputs of projects supported by the Veterinary Directorate, Ministry of Agriculture and Environment Protection, Republic of Serbia (http://www.minpolj.gov.rs/), and Provincial Secretariat for Science and Technological Development, the Autonomous Province of Vojvodina, project no. 114-451-2142/2011 (DP, TP, IHC, AIĆ - http://apv-visokoobrazovanje.vojvodina.gov.rs/). The funders had no role in study design, data collection and analysis, decision to publish, or preparation of the manuscript. MP is affiliated to Avia-GIS NV, which did not play a role in the study design, data collection and analysis, decision to publish, or preparation of the manuscript.

Role of Funders:

Avia-GIS NV provided support in the form of salaries for author (MP) but did not have any additional role in the study design, data collection and analysis, decision to publish, or preparation of the manuscript. The specific roles of the author (MP) are articulated in the ‘author contributions’ section.

Fig 1 copyright:

In compliance with the CC BY 4.0 license, we have changed Fig 1 to include a NaturalEarth baseline map (http://www.naturalearthdata.com/). We have updated the figure caption to include source information.

Authorship changes:

Aleksandra Ignjatović Ćupina (AIC) contributed to the acquisition of the Anopheles hyrcanus data and has been added to the list of authors. All authors are informed and express agreement regarding this change.

Corresponding authorship changes:

Dušan Petrić (DP) is added as the second corresponding author as he was involved in the design and sampling of Anopheles hyrcanus and Culex pipiens.

We appreciate the time and efforts by the editor and advisors in reviewing the manuscript. Please find below detailed responses to the reviewers, whom we thank for their careful consideration of the manuscript. We also reviewed the manuscript for any additional errors and made small changes that are tracked in the attached document (“Revised Manuscript with Track Changes”).

Line numbers for the corrected text are given in red according to the enumeration in the file “Revised Manuscript with Track Changes”.

Reviewer #1: Comments on the manuscript PONE-D-19-16900

Title: Assessment of climate change impact on malaria vectors, West Nile disease, and incidence of melanoma in the Vojvodina Province (Serbia) using data from a regional climate model. Authors: Dragutin T. Mihailović, Dušan Petrić, Tamaš Petrović, Ivana Hrnjaković Cvjetković, Vladimir Đurđević, Emilija Nikolić Đorić, Ilija Arsenić, Mina Petrić, Gordan Mimić

The work presented in paper Mihailović et al. is interesting. The objective of the authors was to compare data from the mosquito field collections and medical studies with regional 29 climate model projections to examine the impact of climate change on the circulation of West Nile virus (WNV), the spreading of the malaria vector Anopheles hyrcanus and the incidence of melanoma. The comparison was done with the coupled regional Eta Belgrade University and Princeton Ocean Model for the period 1961-2015 using the A1B scenario, and the expected changes up to 2030. Overall, significant correlation was found between the frequency of WNV in Culex pipiens and the overwintering temperature averages and seasonal relative humidity at the sampling sites. Correlation was also found between the spreading and relative abundance of Anopheles hyrcanus and the trend of the mean annual temperature. There was also an increase in melanoma incidence.

Minor comments to authors

Title

Authors wrote “malaria vectors” but the only presented data on only one vector Anopheles hyrcanus

Corrected according to the suggestion.

Abstract

L32: ……………and 10-years. Delete the hyphen

L36: Corrected according to the suggestion.

L37: ……………Culex. pipiens. Delete the dot

L44: We abbreviated the genus name to Cx. because name was spelled in the previous sentence.

Introduction

L49-53: Are the authors referring to themselves when they stated, “The authors (……….), have been working together…….”?

Yes, we tried to make it clear with correction below.

L56: Corrected to: “The authors of the manuscript (……….), have been working together…….”

L72: Authors should write, “Climate change……..” instead of “The climate change……..”

L86: Corrected according to the suggestion.

L75: Author should write, “Melanoma mortality…………..within the period 1985-2004

L89-91: Corrected according to the suggestion.

L78: Authors should define the acronym ENCR, as this is used here for the first time.

L83: … using ENCR data 

L93-94: Changed to: 

… using European Network of Cancer Registries (ENCR) data

L81-82: Authors should use past tense in the sentence “……we compared considerable of previously……”

L97: Corrected according to the suggestion.

L54-55: This sentence is not clear for me. I suggest this: “In this paper, we analysed observed data collected over a period of 31 years……”

L61: Corrected according to the suggestion, “collected” erased because of tautology.

Materials and Methods

L96: Authors should define SRES-A1B scenario for the first time.

The SRES-A1B scenario is defined in the text, and central differences to RCP explained. Due to this, authors think that selection of scenario, to some extent, is irrelevant for the presented results.

L96: … and the period 2001-2030 using the SRES-A1B scenario.

L113-122 Corrected to:

 … and the period 2001-2030 according to the A1B scenario defined in Special Report on Emissions Scenarios (SRES) (Nakićenović and Swart, 2000), from now on SRES-A1B. SRES scenarios, which defined future global greenhouse gases emissions, were extensively used in the Intergovernmental Panel on Climate Change (IPCC) Third, and Fourth Assessment Reports. The main storyline behind the A1B scenario is rapid economic growth, followed by a significant increase in greenhouse gases concentrations in the future. In the Fifth Assessment Report (AR5), the Representative Concentration Pathway (RCP) is introduced, which are possible future concentration pathways without any storyline behind them. Comparing SRES-A1B and RCPs in terms of the greenhouse gases concentrations, at the end of this century SRES-A1B is the closest to RCP6.0, but for the time horizon used in this study, up to 2030, the difference between any SRES or RCPs are relatively small. 

New references included:

Summary for Policy makers. In: Nakićenović N, Swart R, editors. Special Report on Emissions Scenarios. Cambridge: Cambridge University Press - Published for the Intergovernmental Panel on Climate Change. 2000. pp. 1-21

L115: Authors used only one formula, the subtitle should be in singular

L141: Corrected according to the suggestion.

Results

- Authors should specify the exact p-values instead of writing p<0.05 or p>0.05

L260-281: Corrected according to the suggestion. The section was updated to include the exact p-values.

L207: “The Poisson regression model for the dependence of a number of detections per site (frequency-λ)…………………………is highly significant”. Authors stated it was highly significant, but from my perspective, p<0.05 is not a specific indication of high significance. Could the give the exact value of p?

L282-283: Corrected according to the suggestion. Exact p value is not below 0,01 (it is 0.01393) which is considered as high significance by many authors, so we erased the word “highly”.

- Fig2b and 2c are fuzzy

Thank you for your comment. The figures were reformatted to higher resolution according to PACE.

- Fig4. Colors of fig4c are confusing

Figures 4c has been adapted to have more contrasting colors.

It will be more interesting if the authors used only vector-borne diseases data in this paper.

N.B: Other comments are incorporated in the manuscript

Authors appreciate very much the effort invested in the improvement of the manuscript quality. All suggestions are incorporated into the revised version except one concerning the spelling of NUTS. Nomenclature of territorial units for statistics is originally abbreviated NUTS from the French version (Nomenclature des Unités territoriales statistiques).

Reviewer #2: 

Authors are presenting an interesting paper regarding the effects of climate change in Northern Serbia considering three independent measures: The spread of Anopheles hyrcanus, the presence of West Nile Virus in Culex pipiens, and the incidence of melanoma cases. The paper is interesting, however, discussion should be improved specially on the uncertainty of future predictions since they are using just one climatic model. Further, their results should be stated more carefully since their model rely on assumptions (e.g., manually selected variables) which are also not clearly stated.

Discussion is corrected according to the reviewer’s suggestion:

L345-352: New text added: 

The temperature trend over the period of observations used in this study and for the future time horizon following A1B scenario obtained with the EBU-POM regional climate model is within the multi-model ensemble (MME) spread of regional climate models with similar configuration used in the ENSEMBLES project (van der Linden and Mitchell, 2009). For the period 2001-2030 the temperature change for the region of interest in the EBU-POM integration is 0.75 �C concerning the period 1961-1990 and for the same period ENSEMBLES MME spread range is 0.5-1.5 �C (MEP, 2017). Following this finding, other results presented in this paper that relay on temperature change, can be seen as an estimate that will be within uncertainty related to the future temperature projection.

New references included:

van der Linden P and Mitchell JFB, editors. ENSEMBLES Climate Change and its Impacts. Summary of research and results from the ENSEMBLES project. Exeter: Met Office Hadley Centre;. 2009.:

Ministry of Environmental Protection of the Republic of Serbia. Second National Communication of the Republic of Serbia under the United Nations Framework Convention on Climate Change [Internet]. Belgrade: The Ministry; 2017 Aug. 162p [cited 2019 Sep 10]. Available from: http://www.klimatskepromene.rs/wp-content/uploads/2017/09/SNC_eng.pdf

Major comments:

The paper is showing results in the order: malaria vector, WNV, and melanoma. I suggest following the same order in the abstract.

Changed as suggested.

Authors are using one of the SRES future climatic scenarios; currently the standard for future climate studies are the RCP scenarios. Authors should describe the nature of the SRES-A1B scenario, which is not mentioned in any part of the study. Further, authors should explicitly discuss uncertainty on their predictions since they are not using other scenarios or other climatic models.

Authors addressed this comment in the text corrected. Please check response to the L113-122 and L 345-352 above.

Lines 176-180. There is no discussion or results regarding these sentences. Was the comparison between EBU-POM model and the Republic Hydrometeorological Service of Serbia perfect? What is the implication of this approach on the overall paper?

This is a valuable comment since the information measure(s) is(are) a good indicator of the reliability of model outputs and thus on the overall paper. The increasing complexity of climate models is a growing concern in the modelling community. However, because we invested a serious effort to make our models more “realistic”, we included more parameters and processes. With increasing model complexity, we are less able to manage and understand the model behaviour. Thus, from a user’s perspective, the following question is entirely natural: “How complex model (EBU-POM model in our case) do I need to use to study this problem (assessment of climate change impact on malaria vectors, West Nile disease, and incidence of melanoma in the VPS) with this data set (temperature and/or precipitation)?”. In the revised version, we inserted the additional text.

L229-249: New text added: 

We considered the papers by Mihailović et al. [2,24] in which Kolmogorov complexity measures (Kolmogorov complexity (KC), Kolmogorov complexity spectrum KC spectrum) and the highest value of the KC spectrum (KCM)) and sample entropy (SE) [25] were used to quantify the regularity and complexity of air temperature and precipitation time series, obtained by the EBU-POM model, representing both deterministic chaos and stochastic processes. We considered the complexity of the EBU-POM model using the observed and modelled time series of temperature and precipitation. We computed the KC spectrum, KC, KCM and SE values for temperature and precipitation. The calculations were performed for the entire time interval 1961–1990: (1) on a daily basis with a size of N =10958 samples for temperature and (2) on a monthly basis with a size N =360 for the precipitation. The simulated time series of temperature and precipitation were obtained by the EBU-POM model for the given period. The observed time series of temperature and precipitations for two stations: Sombor (SO) (88 m a.s.l.) and Novi Sad (NS) (84 m a.s.l.) in the considered area, were taken from daily meteorological reports of the Republic Hydrometeorological Service of Serbia. For both sites, the modelled complexity is lower than the observed one, but with the reliability which is in the interval values allowed by the information measures (KC, KCM, and SE) (Krzic et al. 2011, Dell’ Aquila et al. 2016, Cavicchia et al. 2016). These findings mean that the models with a KC (and KCM) complexity lower than the measured time series complexity cannot always reconstruct some of the structures contained in the observed data. However, it does not mean that outputs from EBU-POM model do not correctly simulate climate elements since both sites values indicate the absence of stochastic influences, providing reliable projections of the climate elements.

New references included:

Krzic A, Tosic I, Djurdjevic V, Veljovic K, Rajkovic B. Changes in some indices over Serbia according to the SRES A1B and A2 scenarios. Climate Research. 2011;49: 73-86. 

Cavicchia L, Scoccimarro E, Gualdi S, Marson P, Ahrens B, Berthou S, et al. Mediterranean extreme precipitation: a multi-model assessment. Clim. Dyn. 2016. doi: 10.1007/s00382-016-3245-x

Dell’ Aquila A, Mariotti A, Bastin S, Calmanti S, Cavicchia L, Deque M, et al. Evaluation of simulated decadal variations over the Euro-Mediterranean region from ENSEMBLES to Med-CORDEX. Clim. Dyn. 2016. doi:10.1007/s00382-016-3143-2

Line 277-280: There is no evidence in this paper supporting this affirmation since the variables analyzed corresponded to three temperature related variables and just one considering humidity. Moreover, results were never compared statistically; modify accordingly.

Corrections made as suggested. The sentence “It seems that temperature in semi-urban areas dominates the other environmental factors influencing WNV circulation in nature (e.g. landscape suitability for reservoir host and mosquito vector, host availability, precipitation), as it is the primary factor affecting both mosquito vector abundance and virus replication.” now reads as:

L400-404: Corrected to: It seems that temperature in semi-urban areas is an essential environmental factor influencing WNV circulation (landscape suitability for reservoir host and mosquito vector, host availability, and precipitation/water availability are somewhat similar in investigated semi-urban areas of VPS), as it affects both mosquito vector abundance and virus replication.

Figure 2: Expand the acronym CRCM. Also, double check the legend, which is describing red and green colors but the figure is only showing different shades of blue.

Corrections made as suggested.

L173-179: Fig 2. (a) The regional climate model EBU-POM projection of the mean annual air temperature (Ta) for the period 1985 - 2030 and: i) number of specimens sampled in one trap during single sampling period (blue columns); ii) the number of sites invaded by An. hyrcanus (light blue columns); and iii) relative number of positive samplings per year (dark blue columns), (b) projected increase in the number of sites invaded by An. hyrcanus (the period 2001-2030 ±S.E.), and (c) projected increase in the number of the specimens sampled in one trap during single sampling period (2001 - 2030 ±S.E.).

Figure 4: Add WD and HD to the corresponding legend of the graphic. Is there a Croatian sentence in the legend? Please describe how the melanoma incidence was calculated, is the y axis showing incidence or number of cases? Cumulative incidence is known to over-represent trends (see reference: Vandenbroucke & Pearce, 2012, doi: 10.1093/ije/dys142), try to use incidence rate instead.

This is a keystone issue in this field of epidemiology. However, it is still under a broad umbrella of discussion. In particularly mentioned reference (Vandenbroucke & Pearce, 2012) the Authors comprehensively considered the place of incidence rates in dynamic populations as well as the cumulative incidence (risk or portion) from an epistemological point of view and also giving very illustrative (educational) examples. Many authors were arguing with some ideas explicated in this paper, also considering some examples. We agree with V&P ideas, but we did not find the place where they explicitly say that it would always be using the incidence rate instead of cumulative (the question of overestimation). To our understanding, they left the space for a situation when the use of cumulative incidence gives acceptable results. For example, it is partly seen in the paper by Wu et al. (2014). There is another moment why we used cumulative incidence. It is well-known that there is a high correlation between sun exposure (and received cumulated doses of the UV radiation) and melanoma. If that doses (or any climate element) on a daily basis are used from regional climate models, they cannot be directly correlated with daily or monthly measured or calculated biological quantities. The reason for that is the fact that from regional climate models, we can estimate just the trend of the considered physical quantity (in our case -UV doses through their cumulative values). Correspondingly it is correlated with cumulative incidence. Having said that, after the end of the statement in Line 336, we inserted the following text.

The legend in Fig. 4(c) and y-axis in Fig 4 (d) are changed as suggested. 

The M&M - Melanoma incidence and UVR was amended by the following text:

L216-221: New text added: 

In the analysis we have used two indicators: (i) melanoma incidence rate that is a measure of the number of new cases ("incidence") per unit of time ("rate") and (ii) cumulative incidence (“incidence proportion” that measures the number of new cases per person in the population over a defined period of time – often called risk or proportion). Melanoma incidence rate (per 100,000 people) for ten years 1995 - 2004 was based on the data obtained from the paper by Jovanović et al. [7]. From these data, we calculated the cumulative incidence. 

The discussion was also amended by the following text:

L437-441: New text added: 

In a cohort study, Wu et al. (2014) considered the impact of long-term UV radiation flux on skin cancer risk. Comparing with participants in the lowest quintile of cumulative UV radiation flux in adulthood, they found that participants in the highest quintile had multivariable-adjusted risks (cumulative incidence). According to Vandenbroucke and Pearce (2012), some studies where cumulative incidence is used can over-represent the trends.

New references included:

Wu S, Han J, Laden F, Qureshi AA. Long-term ultraviolet flux, other potential risk factors, and skin cancer risk: a cohort study. Cancer epidemiology, biomarkers and prevention: a publication of the American Association for Cancer Research, cosponsored by the American Society of Preventive Oncology. 2014;23(6):1080–1089. 

Vandenbroucke JP, Pearce N. Incidence rates in dynamic populations. Int J Epidemiol. 2012;41(5):1472–1479.

In table S3 consider adding the number of mosquito samples per site.

Number of samples added in the table.

Figure 6 can be replaced with the statistics of such graphic for readers’ interpretation.

L442-446: Corrections made as suggested. Figure 6 deleted, the paragraph now reads as:

From a statistical point of view, the linear regression model for modelling the cumulative incidence of melanoma versus the difference of the cumulative UVR doses for hot and warm days (Fig 4d) is acceptable. Parameters are statistically highly significant (r = 0.9712 and p = 0.000003) while analysis of residual distribution shows a good agreement with the normal distribution (Shapiro-Wilk test, W = 0.9608, p = 0.7952).

Authors are justifying the paper under the ‘One Health’ concept, however they are not discussing the idea further. I would like discussing explicitly the benefits of putting together a set of multidisciplinary specialists to the development of the manuscript and how this contribution is part of the one health concept. 

The discussion was amended by the following text:

L322-344: New text added:

Despite globalisation trends, researchers have become "closed" in their ever-smaller communication circles which are not limited by state or national borders but by the professional language and way of thinking. Thus, by the end of the 20th century, the scientific community has been faced with problems in communication within its confines. One of the principal reasons why vector-borne diseases (VBD) are so difficult to predict, is the complex interaction of multiple factors (vector, host, pathogen, environment including short-term weather patterns and long-term climate change) in space and time (Moore 2008, Zimmerman 2014). Only groups from multiple sectors that communicate and work together on specific aspects of VBD systems will be able to answer the most exciting and pressing problems in the field (Moore 2008). Authors of this paper started collaboration in 2003 comparing the climates of the foci of WNV circulation in USA (California Central Valley) and Europe (Bucharest area) with VPS. As compared climates showed quite similar patterns, colleagues from public health and veterinary joined the initial group of meteorologists and medical entomologists. With the idea to better draw upon the resources and insights of the various sectors we designed and implemented research and programmes to achieve better outcomes in the control of zoonoses (diseases that can spread between animals and humans, e.g., WNV disease). This led us to the following achievements: (i) the first detection of WNV in horses in Serbia in 2009 (Lupulović 2011); (ii) the first detection of WNV in mosquitoes in Serbia in 2010 (26); (iii) the first detection of WNV in wild birds in Serbia in 2012 (Petrović 2013); (iv) development and implementation of the national programme of WNV surveillance in mosquito, bird and horse population [8], combined with human surveillance in VPS from 2014; (v) increased visibility to ECDC, EFSA and WHO; (vi) the first detection of imported dengue human case in Serbia in 2016 (Petrović 2016); and (vii) development and implementation of “One Health” programme in VPS from 2018. We are quite sure that much less would have been achieved without multidisciplinary communication and collaboration initiated in 2003, and this paper would not have been compiled.

New references included:

Moore CG. Interdisciplinary research in the ecology of vector-borne diseases: Opportunities and needs. Journal of Vector Ecology 2008;33(2):218–224.

Zimmerman B. Engaging with Complexity: Thrive! A Plan for a Healthier Nova Scotia. 2014; [e-print] Available from: https://thrive.novascotia.ca/sites/default/files/Thrive-Summit-2014-Brenda-Zimmerman-En.pdf.

Petrić D, Hrnjaković-Cvjetković I, Radovanov J, Cvjetković D, Jerant-Patić V, Milošević V, et al. West Nile virus surveillance in humans and mosquitoes and detection of cell fusing agent virus in Vojvodina Province (Serbia). HealthMED 2012;6(2):462–68.

Lupulović D, Martin-Acebes MA, Lazić S, Alonso-Padilla J, Blazquez AB, Escribano-Romero E, et al. First serological evidence of West Nile virus activity in horses in Serbia. Vector Borne Zoonotic Dis. 2011;11(9):1303–5.

Petrović T, Blazquez AB, Lupulović D, Lazić G, Escribano-Romero E, Fabijan D, et al. Monitoring West Nile virus (WNV) infection in wild birds in Serbia during 2012: First isolation and characterisation of WNV strains from Serbia. Eurosurveillance. 2013;18(44):1–8.

Petrović V, Turkulov V, Ilić S, Milošević V, Petrović M, Petrić D, et al. First report of imported case of dengue fever in Republic of Serbia. Vol. 14, Travel Medicine and Infectious Disease. 2016. p. 60–1

Minor comments:

Please use Oxford comma across the manuscript: e.g., Line 30: ‘the malaria vector, and the incidence of melanoma’.

Corrections made as suggested.

Line 28: Authors never discuss problems related with animal health, thus, I suggest avoiding this kind of affirmations (e.g., line 81). 

The reviewer is right, we did not, but we think it is vital to mention animals because WNV is the important zoonotic diseases. Therefore, we would like to include new paragraphs in Introduction and Discussion. 

The introduction was amended by the following text:

L76-80: New text added: 

In Europe, the total number of reported autochthonous WNV infections in 2018 (n=2,083) exceeded, by far, the total number from the previous seven years (n=1,832). During the same transmission season, outbreaks of West Nile fever among equids increased by 30% compared to the number of outbreaks in 2017. In total, 285 outbreaks among equids were reported by the EU Member States in 2018.

New references included:

Epidemiological update: West Nile virus transmission season in Europe, 2018 [Internet]. [cited 2019 Sep 09]. Available from: https://ecdc.europa.eu/en/news-events/epidemiological-update-west-nile-virus-transmission-season-europe-2018

Also, the discussion was amended by following text:

L377-390: New text added:

The WNV transmission cycle involves mosquito vectors and birds, but equines and humans are also susceptible to infection (Kramer et al. 2007, Blitvich 2008). Although WNV infections have been described in a wide variety of vertebrates, birds are the main natural reservoir. Hundreds of wild and domestic avian species have been described as susceptible to WNV infection, but many of these showed only subclinical infection (Komar, 2003). In horses, WNV infection is also frequently clinically unapparent, but around 10% of cases develop neurological disorders with up to 50% mortality rates (Blitvich 2008, Calistri et al. 2010). An increasing number of severe outbreaks in horses have been reported in Europe in the past decade, including a large one that took place in northeast Italy in 2008 involving 251 stables with 794 cases and five deaths (Calistri et al. 2010). From the first detection of WNV in 8 out of 81 found dead wild birds in Serbia (Petrovic et al. 2013), each year WNV nucleic acid was detected in found dead or captured wild birds during summertime (Petrovic et al. 2018). Serological testing of horses sampled during 2009-2010 showed that 12% of 349 horses from the northern part of the Serbia had neutralizing WNV antibodies (Lupulovic et al. 2012). After that, each year horse WNV cases were detected by the positive serological response (IgG and IgM antibody seroconverted horses) (Petrovic et al., 2018) or as a clinical manifestation of West Nile neuroinvasive disease (Medić et al., 2019).

New references included:

Kramer L, Li J, Shi PY. West Nile virus. Lancet Neurol. 2007;6:171–181.

Blitvich BJ. Transmission dynamics and changing epidemiology of West Nile virus. Anim Health Res Rev. 2008;9:71–86.

Komar N. West Nile virus: epidemiology and ecology in North America. Adv Vir Res. 2003;6:185–234.

Calistri P, Giovannini A, Hubalek Z, Ionescu A, Monaco F, Savini G et al. Epidemiology of West Nile in Europe and in the Mediterranean basin. Open Virol J. 2010;4(1):29–37.

Medić S, Lazić S, Petrović T, Petrić D, Samojlović M, Lazić G et al. Evidence of the first clinical case of equine neuroinvasive West Nile Disease in Serbia, 2018. Acta veterinaria 2019;69(1):123–130.

Line 28: Methods on the paper should be written in past tense: e.g., COMPARED. Review this in the rest of the manuscript, e.g., line 82.

Corrections made as suggested.

Line 30: ‘the spreading of ONE malaria vector’

L33-34: Corrections made as suggested.

Line 37: ‘Culex.’ should be corrected, only Cx.?

L44: Corrections made as suggested.

Line 40: This is the first time you are mentioning HD, please expand the acronym, review this in the rest of the manuscript, for example EU in line 50, or ENCR in line 78.

Changed as suggested.

L40: of days with Tmax ≥ 30 �C (HD)

L47: Changed to: of days with Tmax ≥ 30 �C (Hot Days - HD)

L50: … endorsed by the EU 

L58-59: Changed to: … endorsed by the European Union (EU)

L79: … using ENCR data 

L93-94: Changed to: … using European Network of Cancer Registries (ENCR) data

Line 44: Specify the risk that you are addressing with this research.

Changed as suggested.

L51: New text added: … of vector-borne diseases and melanoma.

Line 54: extra ‘Collected’ after ‘observed data’, please erase.

Changed as suggested, the beginning of the sentence now reads as:

L61: In this paper, the authors collected and analysed observed data over a period

Line 55: Add ‘are’ after the word ‘melanoma’ at the end of the sentence.

L63: Changed as suggested.

Line 58: Here you need a reference for the environmental threat represented for the animal and humans at the Pannonian plane. 

L65: References included as suggested.

Line 63: You need a reference for the affirmation of malaria as worldwide detrimental vector-borne disease.

L72: New reference added as suggested.

World Health Organization, World Malaria Report 2018. Geneva: The Organization; 2018. 

Line 70: Consider adding a reference of how temperature and relative humidity are principal abiotic factors for WNV and An. hyrcanus.

Not sure how to respond to this comment. However, references concerning the vector-borne disease and mosquito vector mentioned in the sentence are already given in the text - [10,11]. 

Line 76: Be consistent across the whole manuscript, use either - or – without spaces to separate year timeframes, 1976–2004 is preferred.

Changed as suggested.

Line 102: Add corresponding reference for the Köpen classification.

L128: New reference added as suggested.

Kottek M, Grieser J, Beck C, Rudolf B, Rubel F. World Map of the Köppen-Geiger climate classification updated. Meteorologische Zeitschrift. 2006;15(3):259–263. 

Line 122: “Data were...”

L148: Changed as suggested.

Line 126: 1985–1986.

L152-153: Changed as suggested.

Line 158: Briefly describe the method of WNV detection, i.e., RT-PCR or the corresponding one before referencing Petrovic et al 2018.

L193-199: New text added:

Mosquito pools were tested for WNV RNA presence by TaqMan-based one-step reverse transcription real-time PCR (RT-qPCR) that amplified both lineage 1 and 2 strains. Viral RNA was extracted using the commercial ISOLATE II RNA Mini Kit (Bioline, The Netherlands) according to the manufacturer's instruction. One-step RT-qPCR was conducted using the commercial kit RNA UltraSense™ One-Step qRT-PCR System (Life Technologies Corporation) with the primers and probe that targeted the nucleocapsid protein C gene regions of WNV, as described by Petrović et al. (2018).

Line 172: Describe the indicators briefly before referencing Jovanovic et al 2009.

According to the suggestion, the text placed between 172-174 lines is replaced by the following one.

L216-221: New text added: 

In the analysis we have used two indicators: (i) melanoma incidence rate that is a measure of the number of new cases ("incidence") per unit of time ("rate") and (ii) cumulative incidence (“incidence proportion” that measures the number of new cases per person in the population over a defined period of time – often called risk or proportion). Melanoma incidence rate (per 100,000 people) for ten years 1995 - 2004 was based on the data obtained from the paper by Jovanović et al. [14]. From these data, we calculated the cumulative incidence.

Line 227: Is the formula correct: warm days - WD?

L301-302: Changed to: 

air temperature Tmax ≥ 25 �C (Warm Days - WD)

Line 263: Consider changing ‘indicate that the findings supporting’ by ‘support’

L370-374: The sentence was quietly confusing; we rewrote it to read like this:

Positive trends which are present in our observations might indicate that the findings on the negative influence of UVR and blue-light radiation (this radiation has a wavelength between approximately 380 nm and 500 nm; it has a very short wavelength, and so produces a higher amount of energy) on adult mosquitoes under laboratory conditions [38,39] could not be simply translated to the field.

Line 273: Authors are not showing incidence rates, just presence of WNV in mosquitoes.

L396: Changed as suggested. End of sentence now reads as:

… with a higher frequency of WNV presence in mosquitoes.

---

## [Decision Letter · Decision Letter 1]

29 Oct 2019

PONE-D-19-16900R1

Assessment of climate change impact on the malaria vector Anopheles hyrcanus, West Nile disease, and incidence of melanoma in the Vojvodina Province (Serbia) using data from a regional climate model

PLOS ONE

Dear Ms. Petrić,

Thank you very much for submitting your manuscript "Assessment of climate change impact on malaria vectors, West Nile disease, and incidence of melanoma in the Vojvodina Province (Serbia) using data from a regional climate model" (#PONE-D-19-16900R1) for review by PLOS ONE. As with all papers submitted to the journal, your manuscript was fully evaluated by academic editor (myself) and by independent peer reviewers. The reviewers appreciated the attention to an important health topic, but they raised substantial concerns about the paper that must be addressed before this manuscript can be accurately assessed for meeting the PLOS ONE criteria. Therefore, if you feel these issues can be adequately addressed, we invite you to submit a revised version of the manuscript that addresses the points raised during the review process. We can’t, of course, promise publication at that time.

We would appreciate receiving your revised manuscript by Dec 13 2019 11:59PM. To enhance the reproducibility of your results, we recommend that if applicable you deposit your laboratory protocols in protocols.io, where a protocol can be assigned its own identifier (DOI) such that it can be cited independently in the future. For instructions see: http://journals.plos.org/plosone/s/submission-guidelines#loc-laboratory-protocols

We look forward to receiving your revised manuscript.

Kind regards,

Abdallah M. Samy, PhD

Academic Editor

PLOS ONE

**Reviewers' comments:**

Reviewer's Responses to Questions

**Comments to the Author**

1. If the authors have adequately addressed your comments raised in a previous round of review and you feel that this manuscript is now acceptable for publication, you may indicate that here to bypass the “Comments to the Author” section, enter your conflict of interest statement in the “Confidential to Editor” section, and submit your "Accept" recommendation.

Reviewer #1: All comments have been addressed

Reviewer #2: (No Response)

2. Is the manuscript technically sound, and do the data support the conclusions?

Reviewer #1: Yes

Reviewer #2: Partly

3. Has the statistical analysis been performed appropriately and rigorously? 

Reviewer #1: Yes

Reviewer #2: N/A

4. Have the authors made all data underlying the findings in their manuscript fully available?

Reviewer #1: Yes

Reviewer #2: Yes

5. Is the manuscript presented in an intelligible fashion and written in standard English?

Reviewer #1: Yes

Reviewer #2: Yes

6. Review Comments to the Author

Reviewer #1: A part from the last single remark made, that the authors should consider, all my comments have been addressed. From my perspective, this paper is acceptable for publication.

Reviewer #2: Authors present a corrected version of their manuscript. In reference to my suggestions, the majority of them have been addressed appropriately. Some comments among their current version:

The authors have included a measure of uncertainty based on the results of the ENSEMBLES project, that is, based on an extrinsic source of climatic model uncertainty. This is potentially valid, but please be explicit on your approach, that is, discuss the absence of experimentation with other climatic scenarios (e.g., SRES A2, B1, etc) and the decision to not use other climatic models apart from EBU-POM.

In the amended paragraph from lines 345-352, it is not clear if the EBU-POM model integration to 0.75ºC refers to the temperatures for 2001-2030 or 1961-1990, please improve the wording and punctuation.

In the amended paragraph from lines 229-249, please explain clearly the sentence ‘but with the reliability which is in the interval values allowed by the information measures’. Consider either quantitatively adding the values you are referring to, or define reliability and how a simple simulated model is preferred over complex models in order to explain the observed phenomena.

Other comments:

Line 120: Please add a reference to back up your ‘affirmation’ of RCP scenarios without any storyline behind them.

Line 150: Add a comma after ‘In all years,’

Lines 186-187: ‘relative humidity’ should be another item, i.e. (iv), you are treating it like that in the results.

Line 188: Separate ‘2010 to’

Line 219: Please include here the reference of “Vandenbroucke JP, Pearce N. Incidence rates in dynamic populations. Int J Epidemiol. 2012;41(5):1472–1479. “

Line 219: I would suggest also separating periods of time using the ‘en dash’ that is: 2004–2005 instead of 2004-2005. This should be reviewed in the overall paper for consistency, there are places where you indeed use this but then you use a different format (e.g., line 227).

Lines 224-226: I will strongly recommend authors moving the ‘Empirical Formula’ section to this section of the manuscript. It feels natural to introduce both the terminology and the corresponding formulas in the section discussing melanoma; currently the formula is between the description of the regional model and the environmental sampling of mosquitos, which is off topic for that section. Also, here you are introducing another formula derived from the previous one, which will benefit being close together in your narrative.

Line 230: KC spectrum should be in square brackets, is an abbreviation-like expression that you are using inside a parenthesis.

Line 282: Add the word statistically: ‘Toa, which is statistically significant’

Line 283: Please be consistent in the way you define your variables across the manuscript. Sometimes you call overwintering temperature as Toa, others ToA, others you use sub indices, others you use the full letters. Same for the thousands, sometimes you separate them with ',' others no comma is used (e.g., 10,000 vs 10000).

Line 315: Change the wording, ‘mosquito vectors’ to An. hyrcanus, you are not referencing to other vectors in the paper.

Line 326: Add comma after the word ‘predict’.

Line 386-387: Improve the wording; the word ‘found’ should be after the noun. For example: ‘…in 8 out of 81 dead wild birds found in Serbia […]. Each year WNV nucleic acid was detected in dead or captured wild birds found during summer time’. This second ‘found’ could even be eliminated.

Line 398: How did you center the ellipsoids for Fig5? You should describe this either on the main text or on the legend of the figure. From the current version, apparently the ellipsoids were developed considering a cluster of human cases right? This is not clear.

Line 405: Use the word ‘means’ instead of ‘mean’.

Fig 1B: Please consider changing the color ramp of the altitude of the study region, green-red is not color-blind friendly.

7. PLOS authors have the option to publish the peer review history of their article (what does this mean?). If published, this will include your full peer review and any attached files.

**Do you want your identity to be public for this peer review?** For information about this choice, including consent withdrawal, please see our Privacy Policy.

Reviewer #1: Yes: Koffi Mensah Ahadji-Dabla

Reviewer #2: No

---

## [Author Response · Author response to Decision Letter 1]

19 Nov 2019

Dear Dr Samy,

We are pleased to submit the revised version of “Assessment of climate change impact on the malaria vector Anopheles hyrcanus, West Nile disease, and incidence of melanoma in the Vojvodina Province (Serbia) using data from a regional climate model” (#PONE-D-19-16900R2). We appreciate the time and efforts by the editor and advisors in reviewing the manuscript. Please find below detailed responses to the reviewers, whom we thank for their careful consideration of the manuscript. We also reviewed the manuscript for any additional errors and made small changes that are tracked in the attached document (“Revised Manuscript with Track Changes”).

Reviewers’ comments:

Reviewer #1: Apart from the last single remark made, that the authors should consider, all my comments have been addressed. From my perspective, this paper is acceptable for publication.

Corrected, Line 361in the file “Manuscript_R2”: “… expectedthe intensity ...”replaced by “… expected, the intensity ...”

Reviewer #2: Authors present a corrected version of their manuscript. In reference to my suggestions, the majority of them have been addressed appropriately. Some comments among their current version:

The authors have included a measure of uncertainty based on the results of the ENSEMBLES project, that is, based on an extrinsic source of climatic model uncertainty. This is potentially valid, but please be explicit on your approach, that is, discuss the absence of experimentation with other climatic scenarios (e.g., SRES A2, B1, etc) and the decision to not use other climatic models apart from EBU-POM.

In the amended paragraph from lines 345-352, it is not clear if the EBU-POM model integration to 0.75ºC refers to the temperatures for 2001-2030 or 1961-1990, please improve the wording and punctuation.

Response:

Current version: For the period 2001-2030 the temperature change for the region of interest in the EBU-POM integration is 0.75 �C concerning the period 1961-1990 and for the same period ENSEMBLES MME spread range is 0.5-1.5 �C [43]. Following this finding, other results presented in this paper that relay on temperature change, can be seen as an estimate that will be within uncertainty related to the future temperature projection. 

New version: For the region of interest, temperature change for the period 2001-2030 with respect to the period 1961-1990, in the EBU-POM integration is 0.75 �C and in the case of ENSEMBLES MME spread range is 0.5-1.5 �C [43]. Furthermore, over period 2001-2030, uncertainty in temperature projections, related to the scenarios is just small fraction of total uncertainty (Hawkins and Sutton, 2009), due to the fact that there is no significant difference in greenhouse gases concentrations for different SRES scenarios. Consequently, it can be considered that for the period 2001-2030, in case of other scenarios, the estimated range will be similar. Following this findings, even that selection of single model and single scenario does not allow estimate of the full uncertainty in the future, presented results that relay on temperature change, can be seen as plausible future realisation within total uncertainty range, and realistic response of the complex system for further increase in temperature as a forcing driver.

Hawkins E, Sutton R. The Potential to Narrow Uncertainty in Regional Climate Predictions. Bull Amer Meteor. 2009; 90(8): 1095–1108. https://doi.org/10.1175/2009bams2607.1

Also, we included the following sentence at the end of the discussion:

In the future studies, it will be beneficial to introduce results of other regional climate models, such as multi-model ensemble from the CORDEX framework, as a need for better understanding of different uncertainties in the results.

In the amended paragraph from lines 229-249, please explain clearly the sentence ‘but with the reliability which is in the interval values allowed by the information measures’. Consider either quantitatively adding the values you are referring to, or define reliability and how a simple simulated model is preferred over complex models in order to explain the observed phenomena.

Response:

Current version: We considered the papers by Mihailović et al. [2, 28] in which Kolmogorov complexity measures (Kolmogorov complexity (KC), Kolmogorov complexity spectrum KC spectrum) and the highest value of the KC spectrum (KCM)) and sample entropy (SE) [29] were used to quantify the regularity and complexity of air temperature and precipitation time series, obtained by the EBU-POM model, representing both deterministic chaos and stochastic processes. We considered the complexity of the EBU-POM model using the observed and modelled time series of temperature and precipitation. We computed the KC spectrum, KC, KCM and SE values for temperature and precipitation. The calculations were performed for the entire time interval 1961–1990: (i) on a daily basis with a size of N =10958 samples for temperature and (ii) on a monthly basis with a size N =360 for the precipitation. The simulated time series of temperature and precipitation were obtained by the EBU-POM model for the given period. The observed time series of temperature and precipitations for two stations: Sombor (SO) (88 m.a.s.l.) and Novi Sad (NS) (84 m.a.s.l.) in the considered area, were taken from daily meteorological reports of the Republic Hydrometeorological Service of Serbia. For both sites, the modelled complexity is lower than the observed one, but with the reliability which is in the interval values allowed by the information measures (KC, KCM, and SE) [30, 31,32]. These findings mean that the models with a KC (and KCM) complexity lower than the measured time series complexity cannot always reconstruct some of the structures contained in the observed data. However, it does not mean that outputs from EBU-POM model do not correctly simulate climate elements since both sites values indicate the absence of stochastic influences, providing reliable projections of the climate elements.

New version: We considered the papers by Mihailović et al. [2, 28] in which Kolmogorov complexity measures [Kolmogorov complexity (KC), Kolmogorov complexity spectrum (KC spectrum), and the highest value of the KC spectrum (KCM)], and sample entropy (SE) [29] were used to quantify the regularity and complexity of air temperature and precipitation time series, obtained by the EBU-POM model, representing both deterministic chaos and stochastic processes. We considered the complexity of the EBU-POM model using the observed and modelled time series of temperature and precipitation. We computed the KC spectrum, KC, KCM and SE values for temperature and precipitation. The calculations were performed for the entire time interval 1961–1990: (i) on a daily basis with a size of N =10958 samples for temperature and (ii) on a monthly basis with a size N =360 for the precipitation. The simulated time series of temperature and precipitation were obtained by the EBU-POM model for the given period. The observed time series of temperature and precipitations for two stations: Sombor (SO) (88 m.a.s.l.) and Novi Sad (NS) (84 m.a.s.l.) in the considered area, were taken from daily meteorological reports of the Republic Hydrometeorological Service of Serbia. For both sites, the modelled complexity is lower than the observed one, but with the reliability which is in the interval values allowed by the information measures (KC, KCM, and SE) [30, 31, 32]. The term model reliability we have used in the following context. The Lyapunov exponent (LLE) relates to the predictability of measured time series, which includes deterministic chaos as an inherent component. Model predictability is here understood as the degree to which a correct prediction of a system's state can be made either qualitatively or quantitatively. In a stochastic analysis, a random process is considered predictable if it is possible to infer the next state from previous observations. In many models, however, randomness is a phenomenon which “spoils” predictability (Mihailović, 2019). Deterministic chaos does not mechanically denote total predictability but means that at least it improves the prognostic power. In contrast, stochastic trajectories cannot be projected into the future. If LLE>1 then time series is not chaotic, but is rather stochastic, and predictions cannot be based on chaos theory. However, if 0<LLE<1 it indicatesthe existence of chaos in time series. In that case, one can compute the approximate time (often called Lyapunov time (LT)) limit for which accurate prediction for a chaotic system is a function of LLE. It designates a period when a specific process (physical, mechanical, hydrological, quantum, or even biological) moves beyond the bounds of precise (or probabilistic) predictability and enters a chaotic mode. According to (Frison and Abarbanel (1997) that time can be calculated as LLE Δt_lyap=1/LLE. If LLE→0, implying that Δt_lyap→∞, then long-term accurate predictions are possible. However, many climate time series are highly complex. Therefore, 〖∆t〗_lyapcan be corrected for randomness in the following way. Similar to 〖∆t〗_lyap we can introduce a randomness time 〖∆t〗_rand=1/KC(in time units, second, hour or day). Henceforth, we shall denote this quantity Kolmogorov time (KT), as it quantifies the period beyond which randomness significantly influences predictability. Then, the Lyapunov time corrected for randomness is defined as[〖0,∆t〗_lyap ]∩[〖0,∆t〗_rand ]. It can be stated that the KT designates the size of the time window within time series where complexity remains nearly unchanged. These findings mean that the models with a KC (and KCM) complexity lower than the measured time series complexity cannot always reconstruct some of the structures contained in the observed data. However, it does not mean that outputs from EBU-POM model do not correctly simulate climate elements since both sites values indicate the absence of stochastic influences, providing reliable projections of the climate elements.

Mihailović DT, Nikolić-Đorić E, Malinović-Milićević S, Singh VP, Mihailović A, Stošić T et al. The Choice of an Appropriate Information Dissimilarity Measure for Hierarchical Clustering of River Streamflow Time Series, Based on Calculated Lyapunov Exponent and Kolmogorov Measures. Entropy. 2019; 21: 215.

Frison TW, Abarbanel HDI. Ocean gravity waves: A nonlinear analysis of observations. Geophys Res. 1997; 102(C1): 1051–1059. DOI:10.1029/96JC02993.

Other comments:

Line 120: Please add a reference to back up your ‘affirmation’ of RCP scenarios without any storyline behind them.

Response:

Current version: In the Fifth Assessment Report (AR5), the Representative Concentration Pathway (RCP) is introduced, which are possible future concentration pathways without any storyline behind them.

New version: In the Fifth Assessment Report (AR5), the Representative Concentration Pathway (RCP) is introduced, which are possible future concentration pathways without any storyline behind them (Moss et al., 2008).

Moss R, Babiker M, Brinkman S, Calvo E, Carter T, Edmonds J et al. Towards new scenarios for analysis of emissions, climate change, impacts, and response strategies. IPCC Expert Meeting Report on New Scenarios. Intergovernmental Panel on Climate Change, Geneva. 2008. pp. 1–132 

Line 150: Add a comma after ‘In all years,’

Changed as suggested.

Lines 186-187: ‘relative humidity’ should be another item, i.e. (iv), you are treating it like that in the results.

Changed as suggested.

Line 188: Separate ‘2010 to’

Changed as suggested.

Line 219: Please include here the reference of “Vandenbroucke JP, Pearce N. Incidence rates in dynamic populations. Int J Epidemiol. 2012;41(5):1472–1479. “

Changed as suggested.

Line 219: I would suggest also separating periods of time using the ‘en dash’ that is: 2004–2005 instead of 2004-2005. This should be reviewed in the overall paper for consistency, there are places where you indeed use this but then you use a different format (e.g., line 227).

Changed as suggested.

Lines 224-226: I will strongly recommend authors moving the ‘Empirical Formula’ section to this section of the manuscript. It feels natural to introduce both the terminology and the corresponding formulas in the section discussing melanoma; currently the formula is between the description of the regional model and the environmental sampling of mosquitos, which is off topic for that section. Also, here you are introducing another formula derived from the previous one, which will benefit being close together in your narrative.

The subtitle “Empiricakl Formula” is erased, the text removed and inserted after “... (Bantaski Karlovac), and BG (Beograd)].”within the section Melanoma Incidence. 

Line 230: KC spectrum should be in square brackets, is an abbreviation-like expression that you are using inside a parenthesis.

Response:

Current version: ... in which Kolmogorov complexity measures (Kolmogorov complexity (KC), Kolmogorov complexity spectrum KC spectrum) and the highest value of the KC spectrum (KCM)) and sample ...

 New version: ... in which Kolmogorov complexity measures [Kolmogorov complexity (KC), Kolmogorov complexity spectrum (KC spectrum), and the highest value of the KC spectrum (KCM)], and sample ...

Line 282: Add the word statistically: ‘Toa, which is statistically significant’

Changed as suggested.

Line 283: Please be consistent in the way you define your variables across the manuscript. Sometimes you call overwintering temperature as Toa, others ToA, others you use sub indices, others you use the full letters. Same for the thousands, sometimes you separate them with ',' others no comma is used (e.g., 10,000 vs 10000).

Changed as suggested.

Line 315: Change the wording, ‘mosquito vectors’ to An. hyrcanus, you are not referencing to other vectors in the paper.

Here we refer also to Culex pipiens. The sentence remained as it is.

Line 326: Add comma after the word ‘predict’.

Changed as suggested.

Line 386-387: Improve the wording; the word ‘found’ should be after the noun. For example: ‘…in 8 out of 81 dead wild birds found in Serbia […]. Each year WNV nucleic acid was detected in dead or captured wild birds found during summer time’. This second ‘found’ could even be eliminated.

Response:

Current version: From the first detection of WNV in 8 out of 81 found dead wild birds in Serbia [40], each year WNV nucleic acid was detected in found dead or captured wild birds during summertime [8].

New version: From the first detection of WNV in 8 out of 81 dead wild birds found in Serbia [40], each year WNV nucleic acid was detected in dead or captured wild birds during summertime [8].

Line 398: How did you center the ellipsoids for Fig5? You should describe this either on the main text or on the legend of the figure. From the current version, apparently the ellipsoids were developed considering a cluster of human cases right? This is not clear.

The figure caption was adapted as suggested: 

Current version: Fig 5. Frequency of sampling of WNV infected mosquitoes (1–5 times, coloured numbers) during the period 2010–2016, superimposed over a cluster of mosquito, bird, horse, and human WNV cases in (a) 2014 and (b) 2015 (modified after Petrić et al. [9]).

New version: Fig 5. Frequency of sampling of WNV infected mosquitoes (1–5 times, coloured numbers) during the period 2010–2016, superimposed over a cluster of mosquito, bird, horse, and human WNV cases in (a) 2014 and (b) 2015. The standard deviation ellipse (1-σ) is centered around the mean coordinates of the total number of mosquito, bird, horse and human WNV cases (modified after Petrić et al. [9]).

Line 405: Use the word ‘means’ instead of ‘mean’.

Changed as suggested.

Fig 1B: Please consider changing the color ramp of the altitude of the study region, green-red is not color-blind friendly.

Adapted.

---

## [Decision Letter · Decision Letter 2]

6 Dec 2019

PONE-D-19-16900R2

Assessment of climate change impact on the malaria vector Anopheles hyrcanus, West Nile disease, and incidence of melanoma in the Vojvodina Province (Serbia) using data from a regional climate model

PLOS ONE

Dear Ms. Petrić,

Thank you very much for submitting your manuscript "Assessment of climate change impact on malaria vectors, West Nile disease, and incidence of melanoma in the Vojvodina Province (Serbia) using data from a regional climate model" (#PONE-D-19-16900R2) for review by PLOS ONE. As with all papers submitted to the journal, your manuscript was fully evaluated by academic editor (myself) and by independent peer reviewers. The reviewers appreciated the attention to an important health topic, but they raised substantial concerns about the paper that must be addressed before this manuscript can be accurately assessed for meeting the PLOS ONE criteria. Therefore, if you feel these issues can be adequately addressed, we invite you to submit a revised version of the manuscript that addresses the points raised during the review process. We can’t, of course, promise publication at that time.

We would appreciate receiving your revised manuscript by Jan 20 2020 11:59PM. To enhance the reproducibility of your results, we recommend that if applicable you deposit your laboratory protocols in protocols.io, where a protocol can be assigned its own identifier (DOI) such that it can be cited independently in the future. For instructions see: http://journals.plos.org/plosone/s/submission-guidelines#loc-laboratory-protocols

We look forward to receiving your revised manuscript.

Kind regards,

Abdallah M. Samy, PhD

Academic Editor

PLOS ONE

**Additional Editor Comments:**

I think we gave the authors several opportunities to address the reviewer comments; this is the third revision. We need to address some more comments before considering a decision on this manuscript. Please consider carefully all comments. Thanks!

**Reviewers' comments:**

Reviewer's Responses to Questions

**Comments to the Author**

1. If the authors have adequately addressed your comments raised in a previous round of review and you feel that this manuscript is now acceptable for publication, you may indicate that here to bypass the “Comments to the Author” section, enter your conflict of interest statement in the “Confidential to Editor” section, and submit your "Accept" recommendation.

Reviewer #2: (No Response)

2. Is the manuscript technically sound, and do the data support the conclusions?

Reviewer #2: Yes

3. Has the statistical analysis been performed appropriately and rigorously? 

Reviewer #2: Yes

4. Have the authors made all data underlying the findings in their manuscript fully available?

Reviewer #2: Yes

5. Is the manuscript presented in an intelligible fashion and written in standard English?

Reviewer #2: Yes

6. Review Comments to the Author

Reviewer #2: The authors have addressed the most worrisome comments. A couple of minor changes are suggested:

Line 35: separate ’10years’

Line 56: add a space after ‘World Health Organization’

Line 66: to say that West Nile virus is one of the most detrimental vector borne diseases worldwide is debatable, potentially, dengue is far more detrimental worldwide. On the contrary, malaria is the most detrimental vector borne disease worldwide. Modify accordingly.

Line 110: ‘…the Representative Concentration Pathways (RCPs) are introduced, which are…’, use the plural form.

Line 161: add a space in ‘Figs2a’

Line 193: space in ‘Fig3’

Line 211: add a comma after ’sites’

Line 213: Change ‘Firstly’ for ‘First’

Line 226-227: Change ’N’ for ’n’ to describe sample size as done previously in the manuscript

Line 262: use the plural form ‘trends’

Lines 291-258: I appreciate the clear explanation of the authors for this part of the manuscript. However, I reviewed previous reviews and noticed that although now all the explanation is clear, no results are presented. Is clear that the authors are comparing EBU-POM model vs. the Republic Hydrometeorological Service of Serbia, but what is the conclusion? Is the model reliable? To use their own terminology: what are the results for all their comparisons: LLE Δtlyap = 1/LLE?, ∆trand = 1/KC?, what is the intersection between 0,Δtlyap and 0,∆trand?. Another way to show this information would be: can the EBU-POM model be described by a deterministic chaotic equation? if so, is this demonstrated by this comparison? How all this is related to the overall manuscript. Does this means that that the regional model can be trusted? All these points are raised between these lines and never discussed again.

Line 362: Anopheles has already been written, change to “An. maculipennis”

Lines 457-459: Please add these lines after the corresponding discussion of environmental models in the discussion section (that is, after line 348).

7. PLOS authors have the option to publish the peer review history of their article (what does this mean?). If published, this will include your full peer review and any attached files.

Reviewer #2: No

---

## [Author Response · Author response to Decision Letter 2]

20 Dec 2019

Journal: PLOS ONE

Manuscript#: PONE-D-19-16900R3

Title of paper: Assessment of climate change impact on the malaria vector Anopheles hyrcanus, West Nile disease, and incidence of melanoma in the Vojvodina Province (Serbia) using data from a regional climate model

Dragutin T. Mihailović1, Dušan Petrić2*, Tamaš Petrović3, Ivana Hrnjaković-Cvjetković4,5, Vladimir Djurdjevic6, Emilija Nikolić-Đorić7, Ilija Arsenić1, Mina Petrić8,9,10*, Gordan Mimić11, Aleksandra Ignjatović-Ćupina2

Dear Dr Samy,

We are pleased to submit the revised version of “Assessment of climate change impact on the malaria vector Anopheles hyrcanus, West Nile disease, and incidence of melanoma in the Vojvodina Province (Serbia) using data from a regional climate model” (#PONE-D-19-16900R3). We appreciate the time and efforts by the editor and advisors in reviewing the manuscript. Please find below detailed responses to the reviewers, whom we thank for their careful consideration of the manuscript. We also reviewed the manuscript for any additional errors and made small changes that are tracked in the attached document (“Revised Manuscript with Track Changes”).

Reviewers’ comments:

Reviewer #2:

Line 35: separate ’10years’ Corrected

Line 56: add a space after ‘World Health Organization’ Corrected

Line 66: to say that West Nile virus is one of the most detrimental vector borne diseases worldwide is debatable, potentially, dengue is far more detrimental worldwide. On the contrary, malaria is the most detrimental vector borne disease worldwide. Modify accordingly. 

Response:

Our intention was to say that West Nile virus disease is one of the most detrimental vector-borne diseases worldwide, but construction of the sentence might indicate to the reader that malaria is the first and WNV disease the second most important.

The text was adapted accordingly. … are vectors of malaria and West Nile virus (WNV) disease, respectively, the two vector-borne diseases distributed worldwide [10,11].

Line 110: ‘…the Representative Concentration Pathways (RCPs) are introduced, which are…’, use the plural form. Corrected

Line 161 (158): add a space in ‘Figs2a’ Corrected

Line 193 (189): space in ‘Fig3’ Corrected

Line 211 (208): add a comma after ’sites’ Corrected

Line 213 (210): Change ‘Firstly’ for ‘First’ Corrected

Line 226-227 (223-224): Change ’N’ for ’n’ to describe sample size as done previously in the manuscript Corrected

Line 262 (261: use the plural form ‘trends’ Corrected

Reviewer #2: Lines 291-258: I appreciate the clear explanation of the authors for this part of the manuscript. However, I reviewed previous reviews and noticed that although now all the explanation is clear, no results are presented. Is clear that the authors are comparing EBU-POM model vs. the Republic Hydrometeorological Service of Serbia, but what is the conclusion? Is the model reliable? To use their own terminology: what are the results for all their comparisons: LLE Δtlyap = 1/LLE?, ∆trand = 1/KC?, what is the intersection between 0,Δtlyap and 0,∆trand?. Another way to show this information would be: can the EBU-POM model be described by a deterministic chaotic equation? if so, is this demonstrated by this comparison? How all this is related to the overall manuscript. Does this means that that the regional model can be trusted? All these points are raised between these lines and never discussed again.

Response:

Thanks a lot for your remarks. They point out on your minutious and patient reading the manuscript. Today, it is rare to find such a collegial and professional trait. After careful reading your comments and also ones from the first round of the reviewing process, I have an impression that “we are (you as the reviewer and I as one of the authors) on the same line having a disconnection just in one point”. It seems that our additional text inserted in the manuscript, can make an impression on the reader which is sublimated in the dictum: “she/he cannot see the forest for the trees”. Maybe, partly you are right since your remark is quite “practical”, while my elaboration is more “theoretical” without clearly emphasizing the touch down. I have tried to connect our half-lines in the aforementioned point.

Reviewer #2: However, I reviewed previous reviews and noticed that although now all the explanation is clear, no results are presented. Is clear that the authors are comparing EBU-POM model vs. the Republic Hydrometeorological Service of Serbia, but what is the conclusion?

Response:

You are right. After this text an additional conclusive statement is needed. In the revised version it is done in the following way: after the last statement in the text (line 258) we inserted the following text (Lines 255-257): “Therefore, the EBU-POM model can be considered as a model having high performances and reliability in projection of temperature and participation, two of the most important elements used in different climate research.”

Reviewer #2: Is the model reliable? To use their own terminology: what are the results for all their comparisons: LLE Δtlyap = 1/LLE?, ∆trand = 1/KC?, what is the intersection between 0,Δtlyap and 0,∆trand?. Another way to show this information would be: can the EBU-POM model be described by a deterministic chaotic equation? If so, is this demonstrated by this comparison? How all this is related to the overall manuscript.

Now we see where a potential source of misunderstanding, which is partly introduced by our omitting a conclusive text about EBU-POM model performances (that is done in the revised version). It was your question about meaning of the term - model reliability in the previous reviewing report. To explain that we inserted the text about Lyapunov and Kolmogorov times as the indicators of the model reliability. Certainly, that EBU-POM model equations satisfy conditions that come from LT and KT, i.e. cannot “take away” system into chaos. Therefore, no necessary tests in this paper are needed. More details about this issue can be found in Mihailović et al. (2014) which is included in the text (Line 255) and the reference list.

Mihailovic DT, Mimic G, Arsenic I. Climate predictions: the chaos and complexity in climate models, Advances in Meteorology. 2014; DOI: 10.1155/2014/878249.

Reviewer #2: Does this means that that the regional model can be trusted? All these points are raised between these lines and never discussed again.

Response:

Certainly, that the EBU-POM model is trustable. Moreover, it can be included in the group of frequently used models.

New version: We considered the papers by Mihailović et al. [2,30] in which Kolmogorov complexity measures [Kolmogorov complexity (KC), Kolmogorov complexity spectrum (KC spectrum), and the highest value of the KC spectrum (KCM)], and sample entropy (SE) [31] were used to quantify the regularity and complexity of air temperature and precipitation time series, obtained by the EBU-POM model, representing both deterministic chaos and stochastic processes. We considered the complexity of the EBU-POM model using the observed and modelled time series of temperature and precipitation. We computed the KC spectrum, KC, KCM and SE values for temperature and precipitation. The calculations were performed for the entire time interval 1961–1990: (i) on a daily basis with a size of n=10,958 samples for temperature and (ii) on a monthly basis with a size N=360 for the precipitation. The simulated time series of temperature and precipitation were obtained by the EBU-POM model for the given period. The observed time series of temperature and precipitations for two stations: Sombor (SO) (88 m.a.s.l.) and Novi Sad (NS) (84 m.a.s.l.) in the considered area, were taken from daily meteorological reports of the Republic Hydrometeorological Service of Serbia. For both sites, the modelled complexity is lower than the observed one, but with the reliability which is in the interval values allowed by the information measures (KC, KCM, and SE) [32,33,34]. The term model reliability we have used in the following context. The Lyapunov exponent (LLE) relates to the predictability of measured time series, which includes deterministic chaos as an inherent component. Model predictability is here understood as the degree to which a correct prediction of a system's state can be made either qualitatively or quantitatively. In a stochastic analysis, a random process is considered predictable if it is possible to infer the next state from previous observations. In many models, however, randomness is a phenomenon which “spoils” predictability [35]. Deterministic chaos does not mechanically denote total predictability but means that at least it improves the prognostic power. In contrast, stochastic trajectories cannot be projected into the future. If LLE>1 then time series is not chaotic, but is rather stochastic, and predictions cannot be based on chaos theory. However, if 0<LLE<1 it indicates the existence of chaos in time series. In that case, one can compute the approximate time [often called Lyapunov time (LT)] limit for which accurate prediction for a chaotic system is a function of LLE. It designates a period when a specific process (physical, mechanical, hydrological, quantum, or even biological) moves beyond the bounds of precise (or probabilistic) predictability and enters a chaotic mode. According to Frison and Abarbanel [36] that time can be calculated as LLE Δt_lyap=1/LLE. If LLE→0, implying that Δt_lyap→∞, then long-term accurate predictions are possible. However, many climate time series are highly complex. Therefore, 〖∆t〗_lyap can be corrected for randomness in the following way. Similar to 〖∆t〗_lyap we can introduce a randomness time 〖∆t〗_rand=1/KC (in time units, second, hour or day). Henceforth, we shall denote this quantity Kolmogorov time (KT), as it quantifies the period beyond which randomness significantly influences predictability. Then, the Lyapunov time corrected for randomness is defined as [〖0,∆t〗_lyap ]∩[〖0,∆t〗_rand ]. It can be stated that the KT designates the size of the time window within time series where complexity remains nearly unchanged. These findings mean that the models with a KC (and KCM) complexity lower than the measured time series complexity cannot always reconstruct some of the structures contained in the observed data. However, it does not mean that outputs from EBU-POM model do not correctly simulate climate elements since both sites’ values indicate the absence of stochastic influences, providing reliable projections of the climate elements (37 -Mihailovic et al. 2014). Therefore, the EBU-POM model can be considered as a model having high performances and reliability in projection of temperature and precipitation, two of the most important elements used in different climate research. 

Line 362: Anopheles has already been written, change to “An. maculipennis” Corrected

Lines 457-459: Please add these lines after the corresponding discussion of environmental models in the discussion section (that is, after line 348). Corrected

---

## [Editor Report · Decision Letter 3]

27 Dec 2019

Assessment of climate change impact on the malaria vector Anopheles hyrcanus, West Nile disease, and incidence of melanoma in the Vojvodina Province (Serbia) using data from a regional climate model

PONE-D-19-16900R3

Dear Dr. Petrić,

We are pleased to inform you that your manuscript has been judged scientifically suitable for publication and will be formally accepted for publication once it complies with all outstanding technical requirements.

With kind regards,

Abdallah M. Samy, PhD

Academic Editor

PLOS ONE

---

## [Editor Report · Acceptance letter]

30 Dec 2019

PONE-D-19-16900R3 

Assessment of climate change impact on the malaria vector Anopheles hyrcanus, West Nile disease, and incidence of melanoma in the Vojvodina Province (Serbia) using data from a regional climate model 

Dear Dr. Petrić:

I am pleased to inform you that your manuscript has been deemed suitable for publication in PLOS ONE. Congratulations! Your manuscript is now with our production department. 

With kind regards,

on behalf of

Dr. Abdallah M. Samy 

Academic Editor

PLOS ONE